# An internal promoter underlies the difference in disease severity between N- and C-terminal truncation mutations of Titin in zebrafish

Jun Zou[1], Diana Tran[1], Mai Baalbaki[1], Ling Fung Tang[1], Annie Poon[1], Angelo Pelonero[1], Erron W Titus[1], Christiana Yuan[1], Chenxu Shi[1], Shruthi Patchava[1], Elizabeth Halper[1], Jasmine Garg[1], Irina Movsesyan[1], Chaoying Yin[2,3], Roland Wu[1,4], Lisa D Wilsbacher[1,4†], Jiandong Liu[2,3], Ronald L Hager[5], Shaun R Coughlin[1,4], Martin Jinek[6], Clive R Pullinger[1,7], John P Kane[1,8], Daniel O Hart[1,8], Pui-Yan Kwok[1,9,10], Rahul C Deo[1,4,10,11]*

[1]Cardiovascular Research Institute, University of California, San Francisco, San Francisco, United States; [2]Department of Pathology and Laboratory Medicine, University of North Carolina at Chapel Hill, Chapel Hill, United States; [3]McAllister Heart Institute, University of North Carolina at Chapel Hill, Chapel Hill, United States; [4]Department of Medicine, University of California, San Francisco, San Francisco, United States; [5]Department of Exercise Sciences, Brigham Young University, Provo, United States; [6]Department of Biochemistry, University of Zurich, Zurich, Switzerland; [7]Department of Physiological Nursing, University of California, San Francisco, San Francisco, United States; [8]Department of Biochemistry and Biophysics, University of California, San Francisco, San Francisco, United States; [9]Department of Dermatology, University of California, San Francisco, San Francisco, United States; [10]Institute for Human Genetics, University of California, San Francisco, United States; [11]California Institute for Quantitative Biosciences, San Francisco, United States

*For correspondence: rahul.deo@ucsf.edu

Present address: †Department of Medicine and Feinberg Cardiovascular Research Institute, Northwestern University Feinberg School of Medicine, United States

Competing interests: The authors declare that no competing interests exist.

**Abstract** Truncating mutations in the giant sarcomeric protein Titin result in dilated cardiomyopathy and skeletal myopathy. The most severely affected dilated cardiomyopathy patients harbor Titin truncations in the C-terminal two-thirds of the protein, suggesting that mutation position might influence disease mechanism. Using CRISPR/Cas9 technology, we generated six zebrafish lines with Titin truncations in the N-terminal and C-terminal regions. Although all exons were constitutive, C-terminal mutations caused severe myopathy whereas N-terminal mutations demonstrated mild phenotypes. Surprisingly, neither mutation type acted as a dominant negative. Instead, we found a conserved internal promoter at the precise position where divergence in disease severity occurs, with the resulting protein product partially rescuing N-terminal truncations. In addition to its clinical implications, our work may shed light on a long-standing mystery regarding the architecture of the sarcomere.

## Introduction

The use of genetics in clinical medicine depends on knowledge that an identified mutation confers risk of disease. Recent guidance issued by the American College of Medical Genetics emphasizes

**eLife digest** The heart is able to beat partly because of a large protein called Titin that helps to give heart muscle its elasticity. Mutations that shorten the gene that encodes Titin can cause part of the heart to become enlarged and weakened, a condition called dilated cardiomyopathy. Some people with shortened copies of this protein have a mild form of cardiomyopathy and are able to lead relatively normal lives. Others develop more severe symptoms that prevent the heart from pumping blood effectively and may even cause the individual to need a heart transplant.

Genetic studies have revealed that mutations that shorten the Titin protein by disrupting the portion of the gene corresponding to the latter two-thirds of the protein (which encodes the so-called "C-terminal" end of the protein) cause more severe symptoms than mutations that occur near the start of the gene. But it is not clear why the location of the mutation matters.

To investigate this problem, Zou et al. used a gene-editing tool called CRISPR to create genetically engineered zebrafish. These fish had mutations at one of six different points in the gene that encodes the zebrafish version of Titin. Just as with humans, mutations near the C-terminal end of the gene caused more severe muscle problems in the fish.

Specifically, Zou et al. found that the worst disease was associated with mutations that occurred at or after a "promoter" region within the gene and near this C-terminal end. Normally, the promoter produces an independent smaller form of the Titin protein, which helps to reduce the severity of muscle problems in zebrafish that have mutations near the start of the gene. However, mutations near the C-terminal end of the gene also damage this smaller form, preventing this failsafe from working, and so lead to more severe symptoms. Zou et al. also found this promoter to be active in both mouse and human hearts.

Future work will focus on learning how this smaller form of Titin works to help muscle develop and withstand stress and determine whether increasing its production can overcome the more severe forms of disease.

that the strongest form of evidence supporting causality of a mutation is whether it results in a truncation of the protein (nonsense, frameshift or canonical splice-site mutations), specifically for genes where loss-of-function mutations are known to cause disease (*Richards et al., 2015*). Truncating mutations within alternatively spliced exons and at the extreme C-terminus of the protein must be interpreted with caution, as they may have little to no impact on protein function. However, the guidelines do not address whether truncating mutations at different positions along the length of the protein might differ in phenotype severity or mechanism of action, a finding that could complicate interpretation of truncation mutations for a broad range of inherited diseases.

Truncations in the giant sarcomeric protein Titin (TTN) reveal patterns that are at odds with this uniform assignment of causality. TTN lies along the length of the sarcomere and is thought to mediate passive tension of the muscle fiber, while also serving as a binding scaffold for a large number of proteins, including actin and myosin (*Lewinter and Granzier, 2013*). Individual regions of TTN are annotated according to their localization to the Z-disc, I-band, A-band, and M-line regions of the sarcomere visualized on immuno-electron micrographs *Fürst et al., 1988*. Truncating and/or missense mutations in TTN result in cardiac (dilated cardiomyopathy [DCM]) and skeletal muscle (myofibrillar myopathy) disease (*Chauveau et al., 2014*). Recent estimates suggest that truncation mutations in TTN may explain as much as 25% of DCM (*Herman et al., 2012*), a disease which affects upwards of 1 in 2500 individuals (*Codd et al., 1989*). Although truncations are found along the length of TTN in DCM patients, in patients with the most severe forms of DCM, truncations reside within the C-terminal two-thirds of the protein (from amino acid 14,760 onwards, *Figure 1A*), specifically in the distal I-band and A-band (we will refer to these as C-terminal truncations) (*Herman et al., 2012*; *Roberts et al., 2015*). Multiple TTN exons are alternatively spliced (*Bang et al., 2001*), with many having little to no inclusion within cardiac isoforms, and splicing has been proposed to at least partially explain the exclusion of N-terminal mutations in patients with end-stage DCM (*Herman et al., 2012*). Nonetheless, over 5500 amino acids in the N-terminal region of the protein are constitutive (percent spliced in or PSI >0.9, *Figure 1B*), implying the importance of other mechanisms.

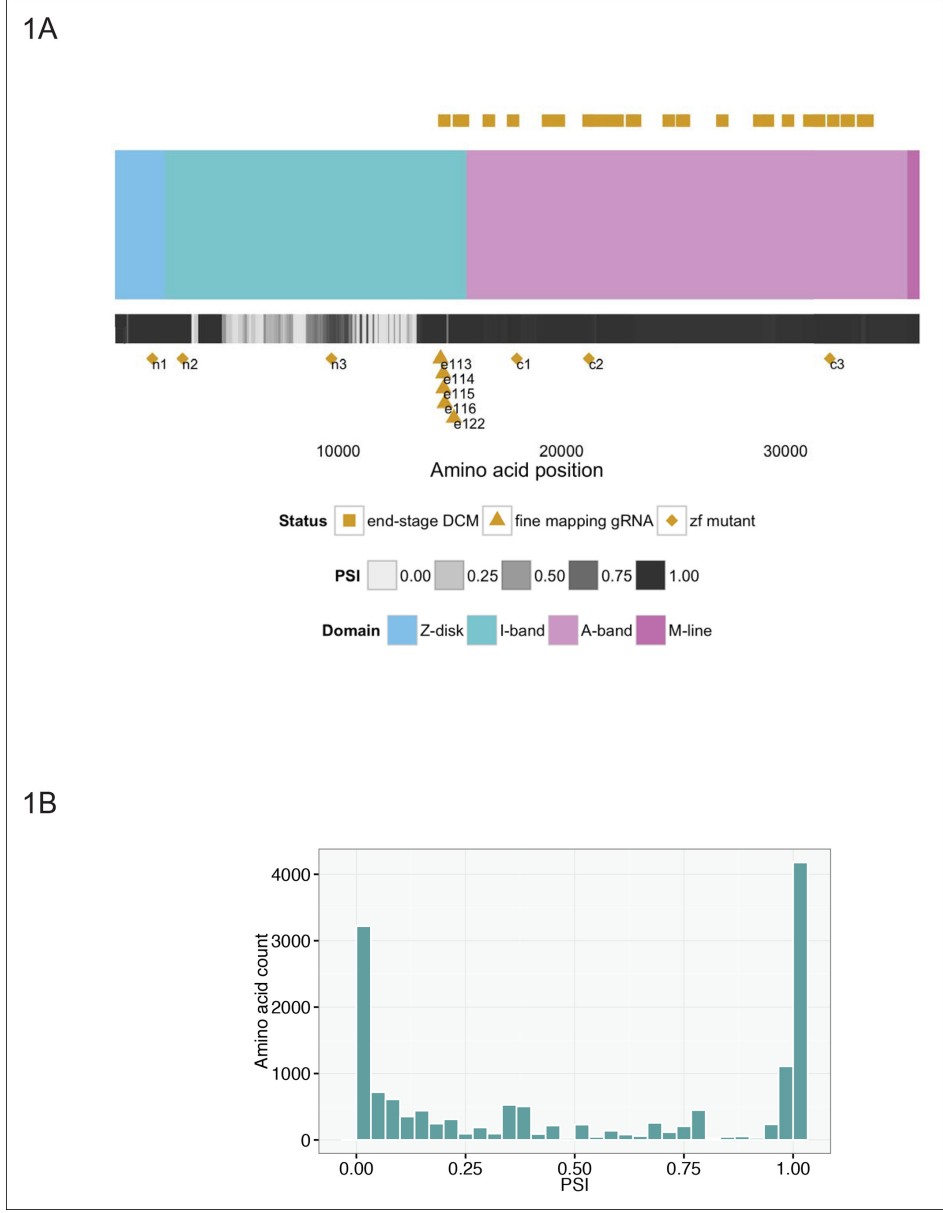

**Figure 1.** Generation of six stable zebrafish lines with truncating mutations in ttna, the zebrafish orthologue of TTN, using CRISPR/Cas9 technology. (**A**) Schematic of the *TTN* meta-transcript (Ensembl ID ENST00000589042), including all known exons with the exception of the Novex-3 exon, which is exclusive to the Novex-3 isoform (Ensembl ID ENST00000360870). Sarcomeric regions are denoted by different colors. A heatmap of PSI values computed from RNA-Seq data of two DCM hearts is shown below the cartoon representation of the transcript. Above the schematic, amino acid positions of truncating mutations of *TTN* in end stage dilated cardiomyopathy identified in a prior study (*Roberts et al., 2015*) are highlighted (gold squares). We restricted our representation to nonsense, frameshift or canonical splice-site mutations and did not include predicted disruptions of putative splice enhancers. A truncating mutation mapping to the Novex-3 exon is not shown. Below the schematic, the corresponding human amino acid positions of zebrafish exons disrupted through CRISPR/Cas9 technology in our work are depicted, including six stable mutant lines (n1, n2, n3, c1, c2, c3, gold diamonds) and individual exons targeted by high dose guide RNA/Cas9 injection (gold triangles). (**B**) Many of the exons not mutated in end-stage DCM are constitutive (i.e. show minimal alternative splicing). Histogram representing the PSI frequency distribution for TTN amino acids N-terminal to amino acid 14,760, which is the most N-terminal amino acid in which a definitive truncation mutation is found in an end-stage DCM patient. The median PSI value was computed for each TTN exon using RNA-Seq datasets from hearts of two patients with DCM (GEO Accession GSM1380722, GSM1380723).

The following figure supplement is available for figure 1:

**Figure supplement 1.** Domain organization of human TTN (top), zebrafish ttna (middle) and zebrafish ttnb (bottom).

# Results

We sought to address the basis of this variation of severity with mutation position. For a disease such as DCM, in which inheritance is most commonly observed as adult-onset autosomal dominant with variable penetrance, deciphering such mechanisms in mouse models can be challenging, as heterozygotes tend to show minimal phenotypes and homozygotes tend to be embryonic lethal (*Gramlich et al., 2009*). We selected the vertebrate zebrafish model for several reasons. Firstly, zebrafish has high sequence conservation of sarcomeric proteins with mammals and recapitulate disease phenotypes with loss-of-function of multiple established Mendelian cardiomyopathy genes (*Dahme et al., 2009*). In fact, forward genetic screens have generated zebrafish *Titin* mutants, although in only one case is the position of the mutation known (*Xu et al., 2002*; *Steffen et al., 2007*; *Myhre et al., 2014*). Secondly, zebrafish can live for up to two weeks with a non-contractile heart, thus allowing mechanistic studies of homozygote mutants. Thirdly, the relatively low cost and large clutch size of zebrafish allow the ability to generate multiple mutant lines and provide statistical power for robust conclusions.

As a result of an ancestral genome duplication event, zebrafish include two titin genes: *ttna* and *ttnb* (*Xu et al., 2002*; *Steffen et al., 2007*; *Seeley et al., 2007*). *Ttna* shares considerable similarity with the human TTN gene (*Figure 1—figure supplement 1*, showing 'meta-transcripts' with all possible exons included). The zebrafish and human proteins are similar in length (~32,000–35,000 amino acids depending on the splice isoform) and share 56% identity and 82% homology. They have a nearly identical domain organization, with three immunoglobulin-like of domains (proximal, mid, and distal), an elastic PEVK region, N2A and N2B domains which distinguish cardiac and skeletal isoforms, a fibronectin-immunoglobulin repeat region that mediates binding to myosin, and a C-terminal kinase domain. In both species, alternative splicing targets the middle Ig-like and PEVK domains, which modulate passive tensile properties of the sarcomere (*Granzier and Labeit, 2004*). Cardiac isoforms typically include much of this elastic region as well as the N2B ± N2A domains, while skeletal isoforms include the N2A domain but exclude N2B as well as much of the elastic region. Multiple mutants isolated from forward genetic screens coupled with morpholino knockdown analysis have demonstrated that *ttna* is essential for cardiac sarcomere formation in zebrafish (*Xu et al., 2002*; *Myhre et al., 2014*; *Seeley et al., 2007*).

Although the paralagous zebrafish *ttnb* gene is dispensable for cardiac sarcomere development (*Steffen et al., 2007*; *Seeley et al., 2007*), it plays an essential role in skeletal muscle sarcomerogenesis. It is expressed at comparable levels to *ttna* in skeletal muscle (as opposed to ~2/3 of *ttna* abundance in heart). In contrast to human TTN or zebrafish ttna, the ttnb protein has abbreviated versions of the middle Ig-like and PEVK elastic domains (*Figure 1—figure supplement 1*) and thus more closely resembles skeletal rather than cardiac muscle isoforms of *ttna*. Morpholino-based skipping of the N2A exon of either *ttna* or *ttnb* results in severe disruption of skeletal muscle sarcomeric architecture, highlighting the mutual contribution of the two genes (*Seeley et al., 2007*).

**Table 1.** Characteristics of six stable zebrafish truncation mutant lines. Size of insertion/deletion, PSI of corresponding exon in embryonic zebrafish heart and skeletal muscle, protein change using standard nomenclature (23), and corresponding human amino acid protein for the *TTN* meta-transcript are shown.

| Mutant ID | Mutation type | PSI of mutated exon (heart/skeletal muscle) | Zebrafish protein change# | Orthologous human amino acid position* |
|---|---|---|---|---|
| n1 | 7bp insertion | 1.000/1.000 | p.E1463fsX3 | 1697 |
| n2 | 1bp insertion | 1.000/1.000 | p.A2815GfsX3 | 3048 |
| n3 | 2bp deletion | 1.000/1.000 | p.Q7597GfsX8 | 9693 |
| c1 | 2bp deletion | 0.996/1.000 | p.E15193GfsX8 | 17996 |
| c2 | 4bp deletion | 1.000/0.995 | p.T18311GfsX8 | 21215 |
| c3 | 8bp insertion | 1.000/0.996 | p.A29052RfsX8 | 32003 |

#Amino acid numbering is relative to ENSDART00000109099. *Amino acid numbering is relative to *TTN* meta-transcript.

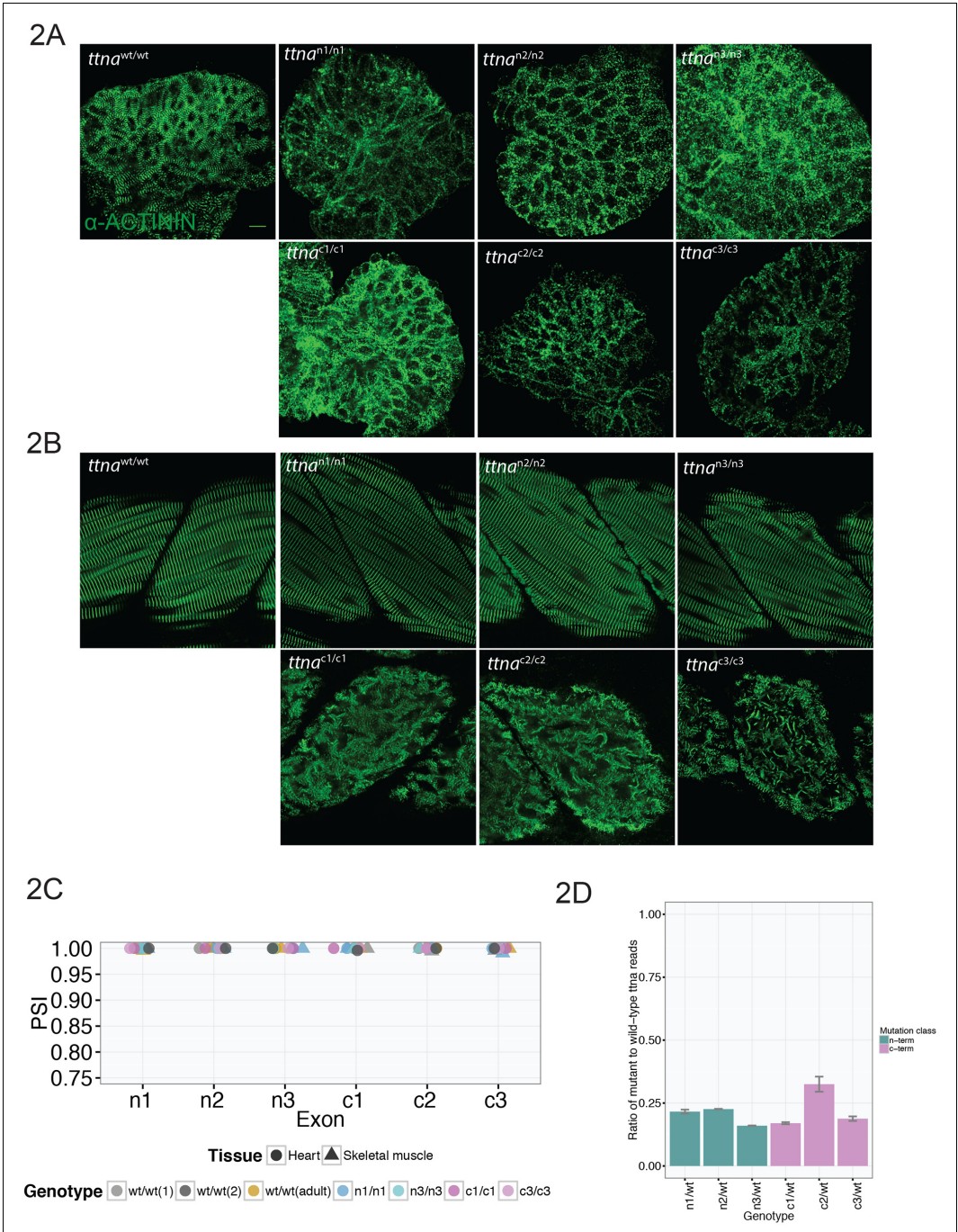

**Figure 2.** C-terminal ttna truncations result in a severe skeletal muscle phenotype while N-terminal truncations are indistinguishable from wild-type. Fixed heart (**A**) and skeletal muscle (**B**) samples of 72 hpf ttna^wt/wt^, ttna^n/n^ and ttna^c/c^ mutant embryos were analyzed by immunostaining for α–actinin, which highlights Z-disc architecture. The cardiac sarcomere was disarrayed in all mutants. However, in skeletal muscle ttna^c/c^ mutants demonstrated severe sarcomeric disarray while ttna^n/n^ mutants retained sarcomeric architecture. Scale bar: 10 uμm. (**C**) All targeted TTN exons are constitutive (i.e. not alternatively spliced) in both cardiac and skeletal muscle. PSI values computed for each mutated exon using RNA-Seq data for dissected hearts and trunk skeletal muscle for various mutant genotypes at 72 hpf. Wild-type fish were analyzed at both 72 hpf and adulthood. Analysis was limited to samples with a sufficient number of exon-exon junction reads to accurately estimate PSI (*Pervouchine et al., 2013*). (**D**) Nonsense-mediated decay reduces mutant transcript levels to ~20–25% of wild-type, but does not vary substantially across mutations. Targeted RNA-Seq was used to determine the ratio of reads derived from the mutant allele vs. the wild-type allele in *ttna^n/wt^*and ttna^c/wt^ heterozygote mutants, which serves as an estimate of nonsense-mediated decay efficiency.

Using CRISPR/Cas9 technology (*Jinek et al., 2012*), we engineered six truncating mutations within the major cardiac TTN orthologue, ttna, (*Figure 1A*, Table 1), so that the resulting protein product would be truncated at either the Z-disk (n1), proximal or mid I-band (n2 and n3) or proximal, mid, or distal A-band (c1, c2, and c3). From wild-type (WT) zebrafish embryo transcriptome data, we anticipated that all exons would be constitutive in both cardiac and skeletal muscle (PSI >0.99, *Table 1*). Mutant lines were bred for two generations to minimize any off-target effects from the genome editing, and we studied the F3 generation for cardiac and skeletal muscle consequences.

As expected (*Xu et al., 2002*; *Gramlich et al., 2015*), ttna truncations showed no gross phenotype in heterozygotes, with all heterozygotes attaining sexual maturity. We thus focused on homozygote mutants, with the expectation that partial rescue from ttnb could potentially elicit phenotypic differences among mutants, especially in skeletal muscle where it is expressed at comparable levels to *ttna* and has a documented essential role. N-terminal (ttna$^{n1/n1}$, ttna$^{n2/n2}$, ttna$^{n3/n3}$, collectively referred to as ttna$^{n/n}$) and C-terminal (ttna$^{c/c}$) truncations all showed severe deficits in cardiac contractility (*Video 1*) and typically died within two weeks of birth. Immunostaining revealed severe cardiac sarcomeric disarray (*Figure 2A*).

In contrast to their cardiac phenotypes, ttna mutants demonstrated a sharp division in development of skeletal muscle disease. All three ttna$^{n/n}$ mutants were capable of grossly normal motion while the ttna$^{c/c}$ truncations were entirely unable to move (*Video 2*). Immunostaining supported the distinction, with ttna$^{c/c}$ mutants having highly distorted sarcomeric architecture while ttna$^{n/n}$ architecture closely resembled wild-type (*Figure 2B*). To confirm transcription of all mutant isoforms, assess nonsense-mediated decay, and rule out selective alternative splicing of ttna$^{n/n}$ mutant exons, we performed RNA-Seq of heart (cardiac muscle) and trunk (skeletal muscle) of 3 day old embryos, as well as adult fish. All mutated exons appeared constitutive in both heart and skeletal muscle, across all genotypes (*Figure 2C*). We next examined the extent of nonsense-mediated decay (NMD) for each mutant allele by performing targeted RNA sequencing of heterozygote embryos and comparing the number of reads for wild-type and mutant exons. In keeping with typical estimates for NMD efficiency (*Zetoune et al., 2008*), we saw ~75–80% of mutant transcript being degraded (*Figure 2D*). However, no substantial variation was seen across the mutants.

Given that variability in alternative splicing or NMD could not explain the selective skeletal muscle disarray in ttna$^{c/c}$ mutants, we hypothesized that this variation in severity could be due to ttna$^{c/c}$ mutants interfering with the action of ttnb, although the ratio of mutant to wild-type transcript of ~20% made this an unlikely scenario. Knockdown of ttnb by morpholino disrupted ttna$^{n/n}$ skeletal muscle morphology indicating that the intact skeletal muscle sarcomeric architecture in ttna$^{n/n}$ mutants arose at least in part from rescue by ttnb protein (*Figure 3A*), although the phenotype was not nearly as severe as that of ttna$^{c/c}$ mutants. To investigate a dominant negative mechanism for

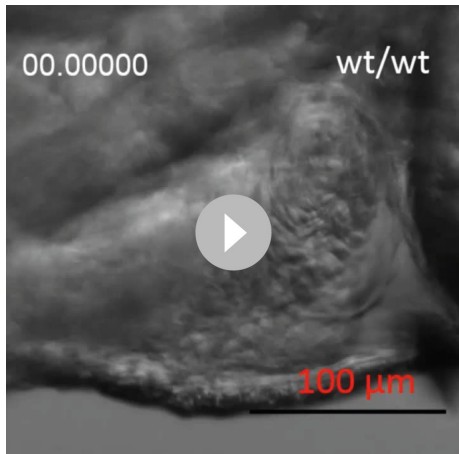

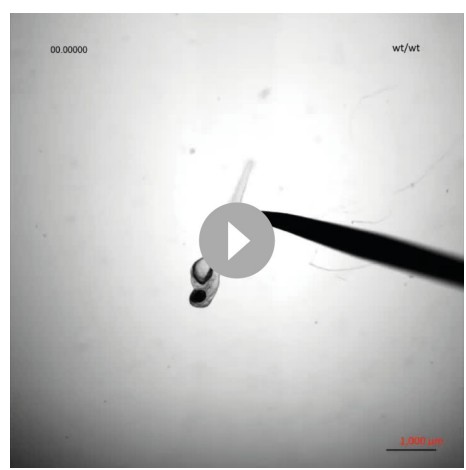

**Video 1.** Cardiac contraction for wild-type, ttna$^{n/n}$ and ttna$^{c/c}$ mutants. All embryos were imaged at 72 hpf.

**Video 2.** Motility assessment for wild-type, ttna$^{n/n}$ and ttna$^{c/c}$ mutants. All embryos were imaged at 72 hpf.

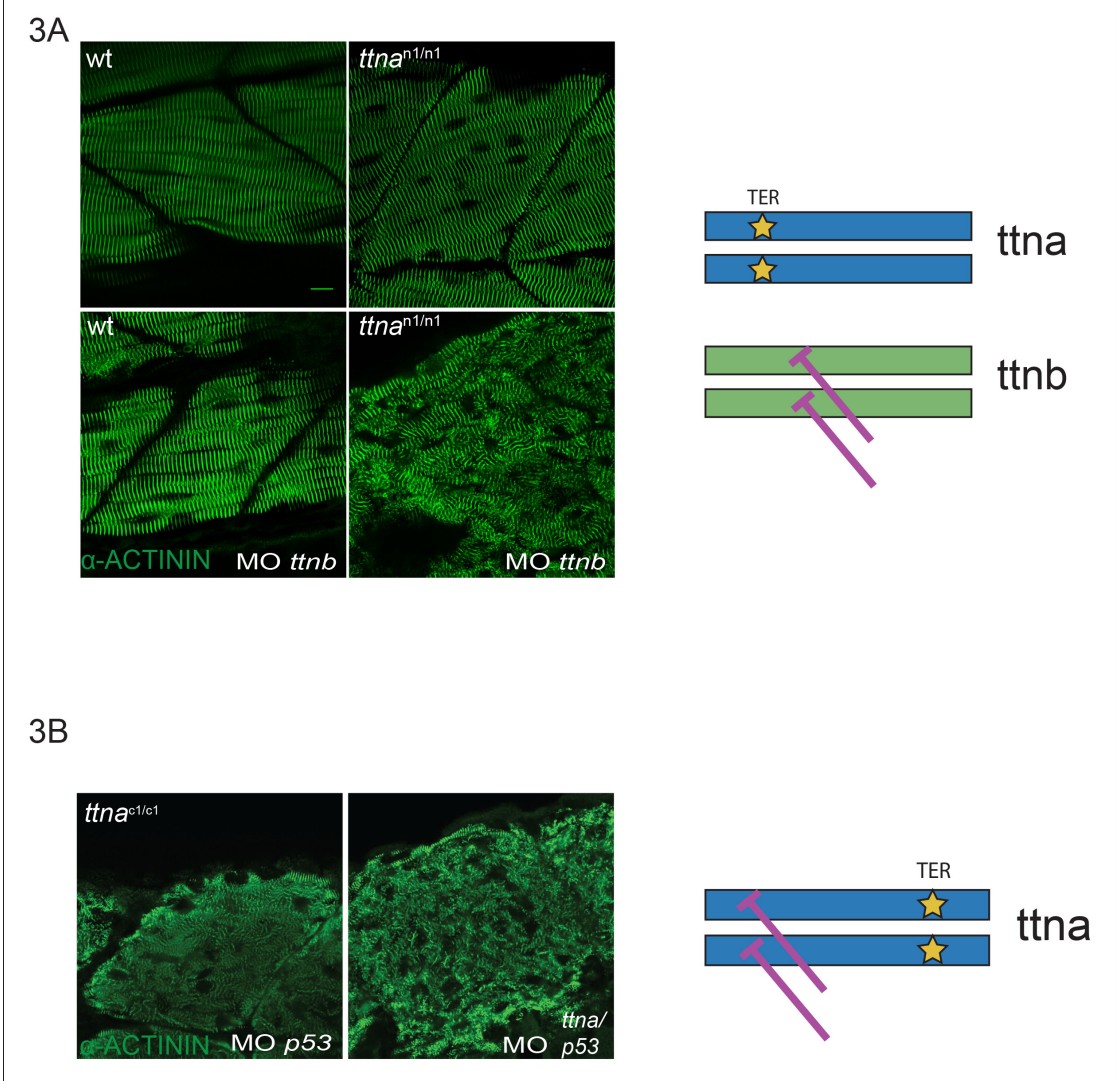

**Figure 3.** C-terminal *ttna* truncations do not act as dominant negatives. (**A**) Knockdown of ttnb by morpholino injection worsens skeletal muscle sarcomeric architecture in ttna$^{n/n}$ mutants. Fixed skeletal muscle samples of 72 hpf ttna$^{n1/n1}$ mutant embryos were analyzed by immunostaining for α–actinin. (right) Cartoon representation of ttna and ttnb proteins, with a premature N-terminal stop codon in ttna (gold star). The purple lines indicate morpholino disruption of the ttnb transcript. Scale bar: 10 µµm. (**B**) Knockdown of ttna by morpholino injection does not rescue skeletal muscle architecture in ttna$^{c1/c1}$ mutants. ttna$^{c1/c1}$ embryos were injected with a ttna splice-site morpholino to the exon 4-intron 4 junction at the 1–2 cell stage and embryos were examined at 72 hpf. At this morpholino dose, knockdown efficacy was estimated at close to 80% and nearly complete cessation of cardiac contraction was achieved in >90% of wild-type embryos (data not shown). Immunostaining for α–actinin revealed no improvement in skeletal muscle architecture. (right) Cartoon representation of mutant ttna proteins, with a premature C-terminal stop codon (gold star). The purple lines indicate disruption of the ttna transcript using a morpholino that is expected to introduce an N-terminal truncation upstream of the C-terminal truncation.

ttna$^{c/c}$ mutants, we performed a morpholino knockdown of ttna, targeting a splice site of an N-terminal constitutive exon (exon 4), which would be expected to introduce an N-terminal frameshift on top of the same C-terminal mutant allele. If the differences between ttna$^{n/n}$ and ttna$^{c/c}$ mutants arose simply from inhibitory effects of the ttna$^{c/c}$ truncated product, then introducing an upstream frameshift should disrupt production of the dominant negative protein product even further and restore sarcomeric architecture. Knockdown by morpholino was effective (~78% knockdown efficiency, data not shown) and the combined action of NMD and morpholino knockdown would be expected to reduce mutant ttna protein to only ~5% of the level of ttnb protein. Nonetheless we were unable to rescue skeletal muscle architecture in ttna$^{c/c}$ mutants (***Figure 3B***). We thus concluded

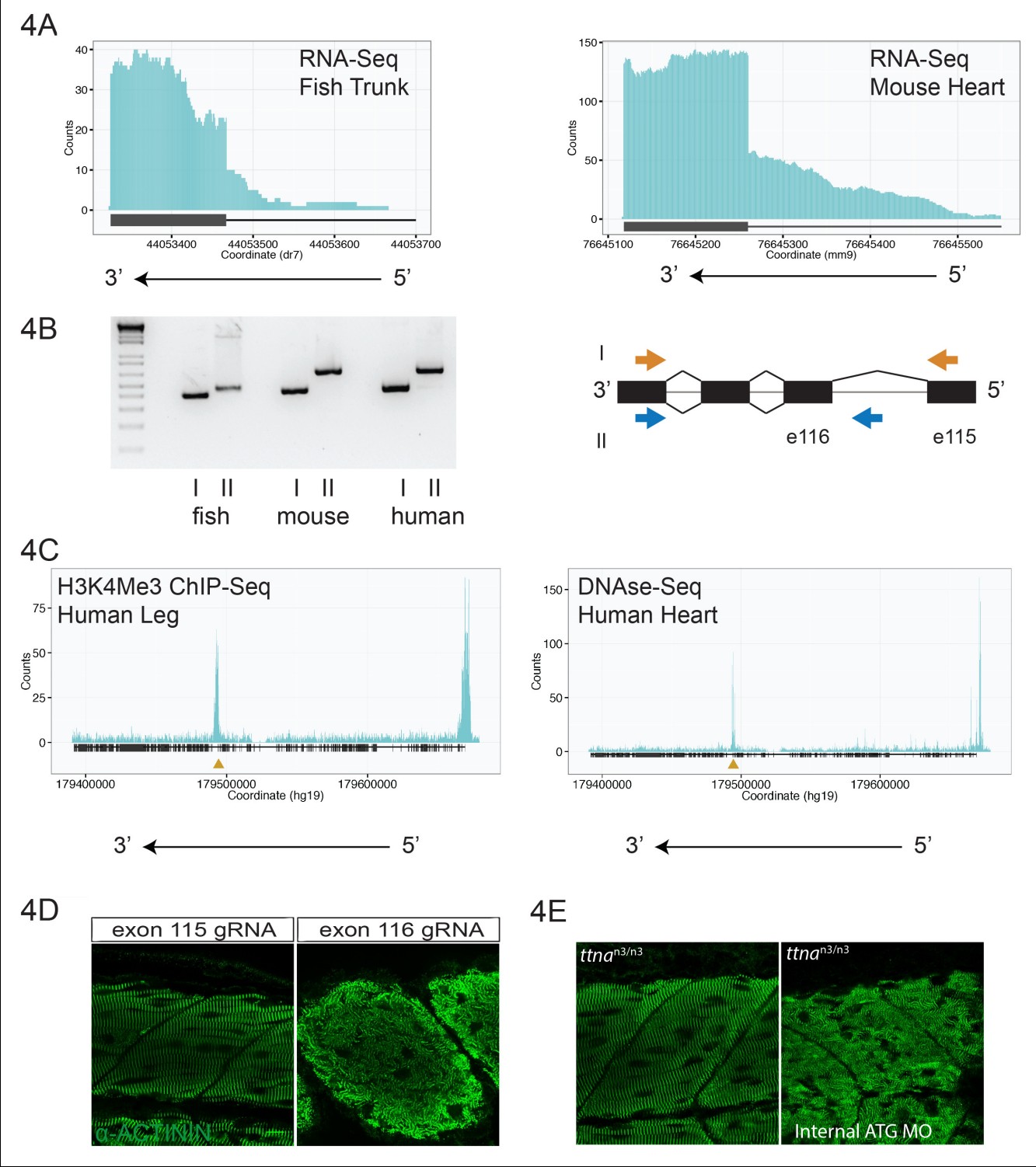

**Figure 4.** An internal promoter at the distal I-band explains phenotypic differences between N- an C-terminal ttna mutants. (**A**) RNA-Seq data from 72 hpf zebrafish trunk (left) and E12.5 mouse hearts (right) depicting accumulation of reads within the intron, upstream of the orthologous exons 116 (zebrafish) and 223 (mouse). The *Ttn* gene is on the negative strand and so reads within the upstream intron are shown to the right of the black bar, which indicates the position of the relevant exon. (**B**) Transcription from an alternative Titin promoter occurs in zebrafish, mouse, and human heart. (left) PCR amplification using a primer in the upstream exon or an internal primer at or near the internal TSS (as determined by 5′-RACE) as forward primer and a shared reverse primer was performed using cDNA from zebrafish, mouse and human heart. For all three species, products of the expected size were found, supporting transcription from an internal promoter. (right) Cartoon representation of PCR amplification scheme, with exon numbering
*Figure 4 continued on next page*

*Figure 4 continued*

according to zebrafish transcript. The zebrafish 5' UTR is shorter than that of mouse and human. (C) (left) H3K4me3 ChIP-Seq data from human fetal leg muscle (GEO Accession GSM1058781), indicating an active promoter overlies exon 240, which is orthologous to exon 116 in zebrafish. The peak at the far right of the figure represents the conventional human *TTN* promoter. (right) DNAse-Seq data from human fetal heart (GEO Accession GSM665830) indicates highly accessible chromatin overlying exon 240, which typically demarcate enhancers or promoters (*Neph et al., 2013*). The accessible chromatin peak at the far right is the conventional TTN promoter. The gold triangle in both panels indicates the genomic position of the human internal promoter TSS identified through 5'-RACE. (D) Phenotypic divergence in skeletal myopathy in *ttna* truncations occurs at the exons flanking the alternative promoter. Cas9 protein and high dose gRNAs corresponding to exon 115 or exon 116 were injected into 1 cell embryos. At 72 hpf, embryos with complete or near complete cessation of cardiac contraction were collected and immunostained for α–actinin to assess skeletal muscle architecture. (E) An ATG morpholino targeting the intronic region upstream of the novel initiator methionine is sufficient to create skeletal muscle disarray on a $ttna^{n3/n3}$ background. 1–2 cell embryos from a $ttna^{n3/wt}$ cross were injected with an intronic morpholino and $ttna^{n3/n3}$ homozygotes immunostained for α–actinin to assess sarcomere architecture in the skeletal muscle. No skeletal muscle disarray was seen in wild-type fish injected with the internal ATG morpholino (data not shown).

The following figure supplements are available for figure 4:

**Figure supplement 1.** H3K4me3 Chip-Seq analysis of human fetal muscle (top, GEO Accession GSM1160200) and DNAse-Seq analysis of fetal human heart (bottom, GEO Accession GSM665817) identifies an internal chromatin accessible peak overlying the alternative TTN promoter.

**Figure supplement 2.** The transition between mild and severe skeletal phenotypes occurs at intron 115, the site of the alternative promoter.

that $ttna^{c/c}$ mutants do not act as dominant negatives and that an alternative explanation was needed.

Another possible explanation for the discrepancy in disease severity between $ttna^{n/n}$ and $ttna^{c/c}$ mutants could be an internal promoter that produces a C-terminal isoform, which could partially rescue N-terminal truncations, especially in conjunction with expression of ttnb. Even though N- and C-terminal truncating mutations would both disrupt full-length ttna, the latter would also disrupt a C-terminal isoform while the former would not. Interestingly invertebrates, including *C. elegans* and *D. melanogaster*, have distinct genes that encode for protein products corresponding to the N-terminal (Z-disc and proximal I-band) or C-terminal (distal I-band and A-band and M-line) portions of Titin (*Bullard et al., 2002*). The Novex-3 isoform in humans (discussed below) does in fact include only the Z-disc and proximal I-band portions of TTN, but no isoform corresponding only to the C-terminal portion of TTN has ever been reported in vertebrates.

Using the exon boundaries defined by $ttna^{n3/n3}$ and $ttna^{c1/c1}$, we examined zebrafish and mouse cardiac and skeletal muscle transcriptomic data for patterns of reads indicating a promoter upstream of one of the ttna exons. We noticed an accumulation of reads 5' to exon 116 (e116, ENS-DART00000109099) in zebrafish skeletal muscle (*Figure 4A*) and in the corresponding exon of mouse heart (e223, ENSMUST00000099981, *Figure 4A*). We hypothesized that these reads arose from transcription from an alternative promoter within the e115-e116 intron. 5'-RACE confirmed an alternative transcription start site (TSS), 110 bp upstream of e116 (within the e115-e116 intron) in zebrafish skeletal muscle, with an in-frame initiator methionine located 7 amino acids upstream of e116. To establish conservation in mammals, we repeated 5'-RACE on adult mouse and fetal human cardiac tissue and found a TSS in the upstream intron (257bp upstream of e223 in mouse; 256bp upstream of the orthologous exon, e240, in humans), with a putative initiator methionine 12 amino acids upstream of the relevant exon. Using PCR, we confirmed the presence of a multi-exonic transcript that includes this alternative promoter site in zebrafish, mouse, and human cardiac tissue (*Figure 4B*).

We next consulted publicly available chromatin accessibility and activation data to find supporting evidence for an internal TTN promoter at this site. DNAse-Seq is a powerful tool to identify regions of accessible chromatin, which typically correspond to promoter or enhancer regions (*Neph et al., 2013*). Examination of multiple fetal human cardiac and skeletal muscle DNA-Seq datasets revealed a prominent peak overlying e240, the orthologous exon to e116 in humans (*Figure 4C*, *Figure 4—figure supplement 1*). A similar peak was found in H3K4me3 Chip-Seq data, a mark for active promoters (*Zhou et al., 2010*), from skeletal muscle and cardiac tissue (*Figure 4C*, *Figure 4—figure supplement 1*). Unbiased epigenetic data thus provides additional strong support for an internal

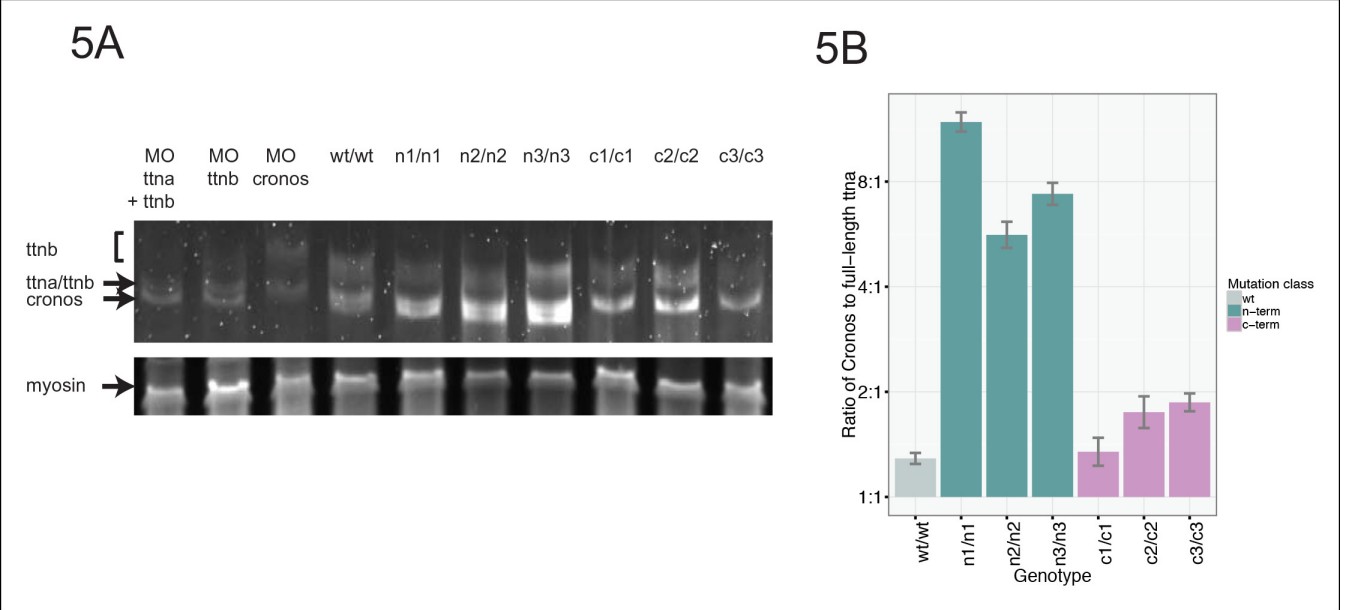

**Figure 5.** Premature termination codons in *ttna* differentially affect the levels of Cronos and full-length *ttna* depending on mutation location. (**A**) 1.0% agarose gel reveals higher molecular weight bands corresponding to different splice isoforms of ttnb (all fish) and ttna (wild-type only). The arrow represents *Cronos* isoform of *ttna*, which is absent in C-terminal truncation mutants and in zebrafish treated with an ATG morpholino against this isoform. Ttna and ttnb expression results in a complex distribution of splice isoforms, whose identity and composition vary over early development (*Steffen et al., 2007*). (**B**) NMD results in a high ratio of *Cronos* to full-length *ttna* transcript levels in N-terminal but not C-terminal truncation mutants. Quantitative real-time PCR was used to estimate relative levels of the *Cronos* and full-length *ttna* trancripts for wild-type and mutant embryos. No efficiency difference was seen for the corresponding primer pairs.

promoter. For convenience, and as a counterpart to the N-terminal Novex-3 isoform, we hereafter refer to this C-terminal Titin isoform as the *Cronos* isoform.

If a protein product arising from the *ttna* internal promoter ameliorates disease severity in $ttna^{n/n}$ but not $ttna^{c/c}$ mutants, we should be able to pinpoint the phenotypic transition at e116 – i.e. truncation mutants N-terminal to e116 should have mild phenotypes while those at or C-terminal to e116 should have severe phenotypes. Interestingly, the corresponding human genomic position (the e239-e240 intron in ENST00000589042) represents the exact transition point for where disease severity markedly increases in DCM TTN truncation patients (*Figure 1A*). We used high dose guide RNA (gRNA) and Cas9 protein injections to exons 113, 114, 115, 116, and 122 and examined cardiac and skeletal muscle architecture and function in the resulting fish, focusing on embryos with nearly complete cessation of cardiac beating, as these would be expected to have a high likelihood of homozygous genomic disruption (given that only homozygote and not heterozygote $ttna^{n/n}$ or $ttna^{c/c}$ mutants have a cardiac phenotype). As expected, e113, e114 and e115 F0 mutants showed intact skeletal muscle architecture while e116 and e122 mutants had severe skeletal muscle disruption (*Figure 4D*, *Figure 4—figure supplement 2*). To confirm that the *Cronos* isoform protein product was actually translated, we used agarose protein electrophoresis and found that a higher mobility (smaller size) protein product was indeed found in wild-type and $ttna^{n/n}$ mutants but not in $ttna^{c/c}$ mutants (*Figure 5A*). Finally, a translation start site morpholino targeted upstream of the actual alternative initiator methionine (and 38 nucleotides upstream of the intron-exon junction) was sufficient to completely impair motility and disrupt skeletal muscle architecture in the $ttna^{n3/n3}$

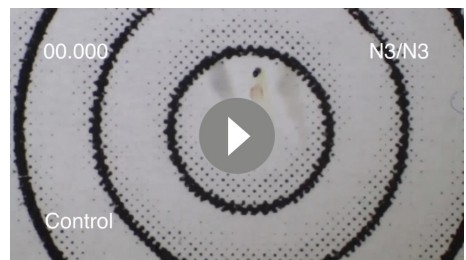

**Video 3.** Motility for wild-type and $ttna^{n/n}$ mutants injected with internal promoter ATG morpholino

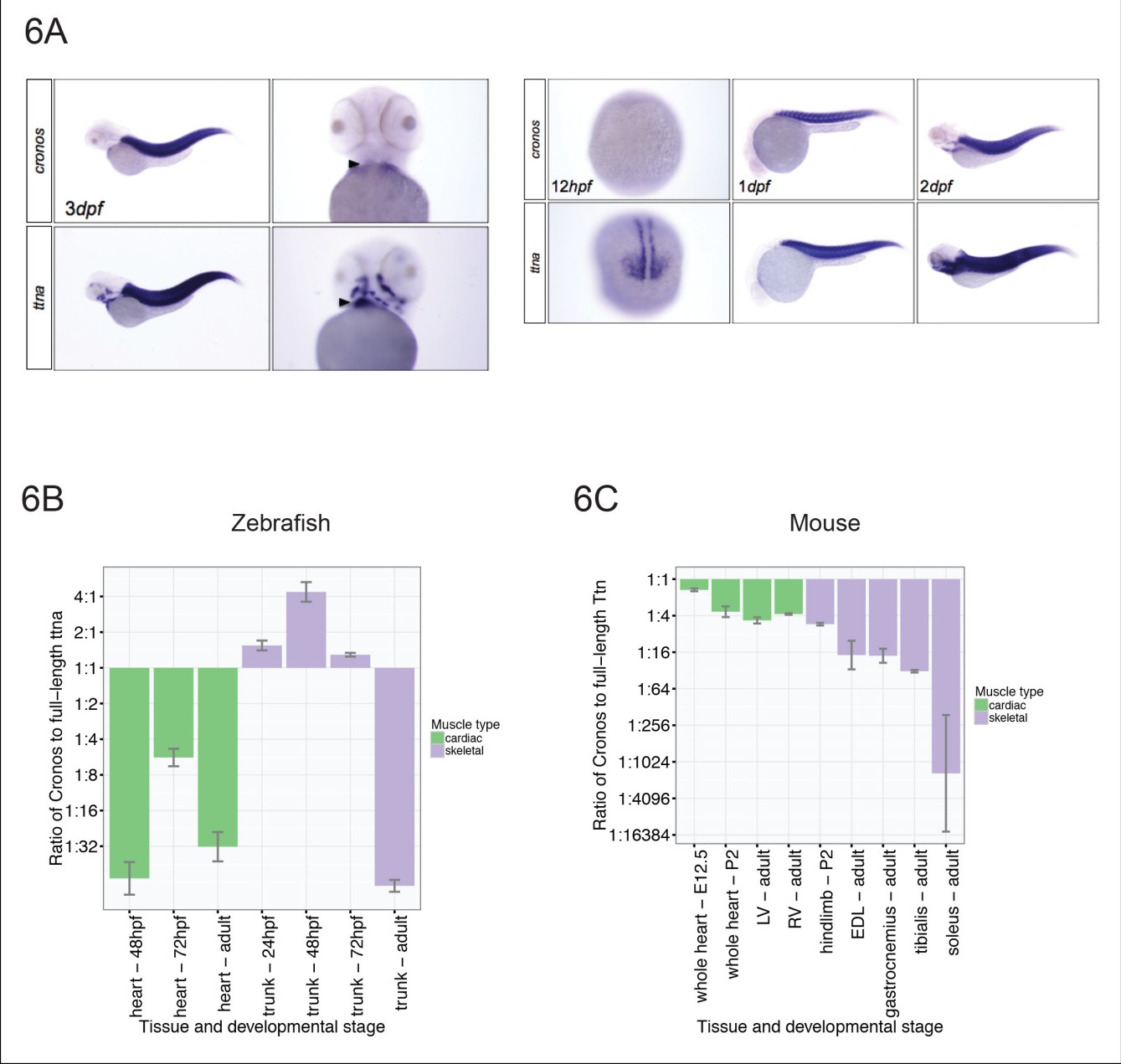

**Figure 6.** Tissue and developmental profile of *Cronos* and *ttna* expression in zebrafish and mouse heart and skeletal muscle. (A) In situ hybridization at 72 hrpf (left) and 24 and 48 hpf (right) reveal prominent expression of both *Cronos* and full length ttna isoforms in zebrafish somites. In contrast, *Cronos* expression in developing zebrafish heart (arrow) is at low levels. (B) Quantitative PCR estimates of the ratio of *Cronos* to full-length ttna reveals prominent expression in developing zebrafish skeletal muscle, but markedly reduced expression in developing heart and in adult heart and skeletal muscle. (C) Quantitative PCR estimates of the ratio of *Cronos* to full-length Ttn reveals prominent expression in developing mouse hearts, which diminishes through development. Skeletal muscle *Cronos* expression is comparable in early postnatal mouse, but diminished in adulthood and varies by skeletal muscle bed. EDL = extensor digitorum longus. LV/RV = left/right ventricle.

mutant (*Figure 4E*, *Video 3*), further supporting the hypothesis that the *Cronos* product can rescue skeletal muscle architecture in ttna[n/n] mutants. No obvious impact of the morpholino on alternative splicing of neighboring exons was seen (data not shown).

Given that NMD substantially reduces the levels of *ttna* transcripts with premature termination codons (*Figure 2D*), we reasoned that we should be able to distinguish N- and C-terminal truncation mutants by the ratio of *Cronos* to full-length transcript. In ttna[n/n] mutants, which would degrade full-

length ttna but not *Cronos*, we would anticipate a higher *Cronos*:full-length ratio than in ttna$^{c/c}$ mutants mutants, which should degrade both transcripts approximately equally. As expected, we see a much higher ratio of *Cronos*:full-length transcript levels in ttna$^{n/n}$ mutants than in ttna$^{c/c}$ mutants (*Figure 5B*).

We next surveyed the expression levels of *Cronos* in cardiac and skeletal muscle in zebrafish at different developmental stages using both in situ hybridization (*Figure 6A*) and real-time PCR (*Figure 6B*). *Cronos* is expressed at high levels in zebrafish skeletal muscle during development (~2:1 ratio of *Cronos* to full-length ttna), but drops off sharply during adulthood (~1:70 ratio). Although the *Cronos* expression in zebrafish heart also appears to be developmentally regulated, it is much lower than in skeletal muscle, with a ~1:5 ratio at 72 hpf and 1:30 ratio in adulthood. This low cardiac *Cronos* expression (coupled with a minor role of *ttnb* in zebrafish hearts) is a likely reason for the severe and indistinguishable cardiac sarcomere disruption in ttna$^{n/n}$ and ttna$^{c/c}$ mutants.

Given that the primary observation that motivated this work was variation in phenotypic severity in cardiomyopathy patients, we next looked to see if there was appreciable *Cronos* expression in mammalian hearts. We found that *Cronos* levels were much higher in developing mouse hearts than in zebrafish, with nearly equal transcript levels of *Cronos* and full-length Ttn (*Figure 6C*) at embryonic day 12.5. Thereafter mouse cardiac expression of *Cronos* decreases to ~20–30% of full-length Titin. Skeletal muscle expression varies across development and in different muscle beds with a 1:5 ratio in mouse hindlimb at P2, and between 1:16 and ~1:1000 ratios in adult extensor digitorum longus (fast-twitch) and soleus (slow-twitch) muscle, respectively.

Although our zebrafish work explains why a milder phenotype would be expected in individuals with truncation mutants upstream of the alternate promoter, it does not answer whether such N-terminal truncations would still result in deleterious manifestations in heterozygote patients. Specifically, an open question remains whether all TTN truncations lead to cardiac or skeletal muscle phenotypes, or conversely, do any TTN truncations lack cardiac or skeletal muscle consequences. In a disease such as dilated cardiomyopathy, with a variable age of onset, it can be challenging to derive general conclusions from observational data of control subjects, many of whom are young or middle-aged. Furthermore, self-reports of 'health' may not reveal limitations in an individual's ability to attain full exercise potential, especially with aging.

To assess whether any *TTN* truncations are compatible with a high degree of cardiovascular fitness throughout life, we turned to a population genetics approach, and sequenced the complete TTN exome (as well as that of 104 other Mendelian cardiac disease genes, *Supplementary file 1A*) in 199 competitive senior athletes who entered as contestants in the Huntsman World Senior Games (median age 73, *Table 2*). We compared the frequency and distribution of TTN truncation mutations to those found in unselected controls and healthy volunteers from a recent sequencing study (CTL$_{lit}$) (*Roberts et al., 2015*). The overall rate of TTN truncations did not differ between senior athletes and CTL$_{lit}$ (1.5 vs 1.2%, p = 0.91). However, the distribution of TTN truncations differed significantly between groups (*Figure 7*, *Figure 7—figure supplement 1*): TTN truncations in senior athletes mapped exclusively to the extreme C-terminal exon of the rare Novex-3 isoform (*Bang et al., 2001*), at a nearly 8-fold higher rate than CTL$_{lit}$ (p = 0.003). This exon is found only in the Novex-3 isoform, which is expressed at low levels in the human heart (<10% of the levels of major TTN isoforms [*Roberts et al., 2015*]), and we would thus expect truncations in this exon would have little effect on TTN function. We next analyzed the ratio of mutants up- and downstream of *Cronos* in affected DCM patients, literature controls and senior athletes (*Figure 7*). As expected, in end-stage DCM we observed a much higher rate of mutants C-terminal to *Cronos* than N-terminal (30:1,

**Table 2.** Demographic characteristics of 199 senior athletes genotyped by capture-based targeted sequencing for this study.

| | |
|---|---|
| Number | 199 |
| Age | 73 (IQR 68-77) |
| Sex | 60% Male |
| Race | 97% European American |
| Sports | Track and field, volleyball, cycling, softball, marathon, . . . |

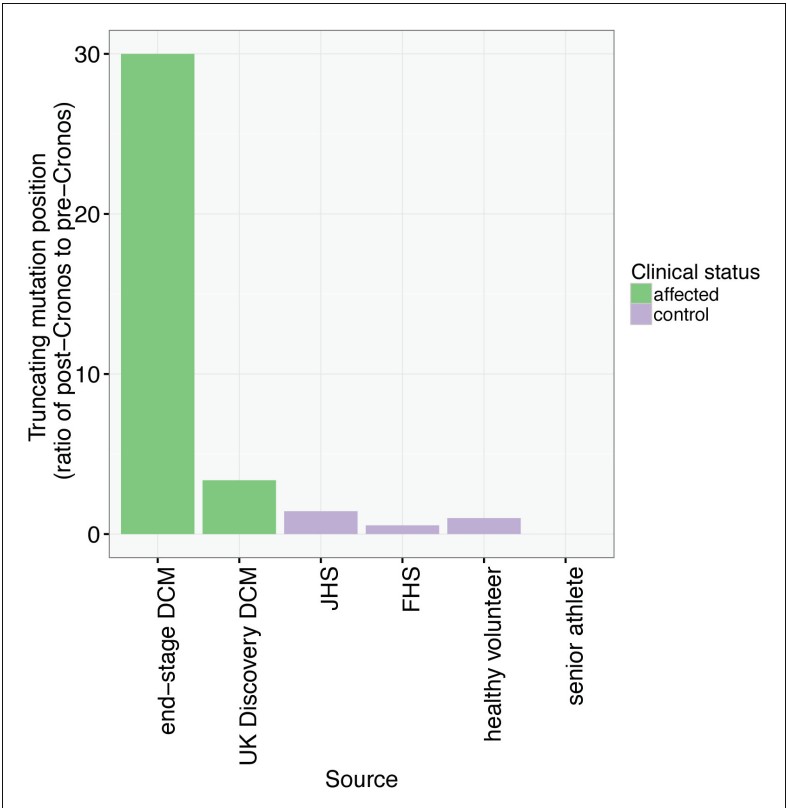

**Figure 7.** TTN truncations are found preferentially upstream of the *Cronos* promoter in controls but downstream of *Cronos* in DCM cases. Ratio of the number of truncating mutations found C-terminal to the position of the *Cronos* promoter to those found N-terminal, in DCM cases and controls derived from the literature (*Roberts et al., 2015*) and in senior athletes.

The following figure supplement is available for figure 7:

**Figure supplement 1.** Mapping of TTN truncation mutations seen in literature cases and controls (*Roberts et al., 2015*) and senior athletes onto a schematic of the TTN meta-transcript.

---

followed by unselected DCM (37:11), literature controls (6:11, 2:2, 10:7 for FHS, healthy volunteers and JHS) and senior athletes (0:3).

## Discussion

Loss of function experiments across a number of systems have demonstrated Titin's importance in diverse aspects of sarcomere development and function, including the ordered assembly of sarcomeric constituents (*Xu et al., 2002*; *van der Ven et al., 2000*), cardiomyocyte systolic (*Hinson et al., 2015*) and diastolic function (*Radke et al., 2007*), and resistance to neurohormonal and vasoconstrictor stresses (*Gramlich et al., 2009*). Our work further contributes the discovery of a conserved internal promoter in *TTN*, whose protein product is essential for skeletal muscle sarcomere development in zebrafish. The *Cronos* promoter is prominently expressed in developing mouse heart, and its location coincides strikingly with the position of TTN truncations in patients with the most severe forms of dilated cardiomyopathy. Our working model is that disruption of both the full-length *TTN* and *Cronos* protein products results in more severe disease than disruption of the full-length product alone.

Throughout this work, we have focused on skeletal muscle architecture and function in zebrafish *ttna* homozygote mutants, given that this tissue demonstrates a sharp and rapidly developing phenotypic distinction between N- and C-terminal Titin truncations. The early onset of this unambiguous phenotypic difference combined with the ease of genetic and transcriptomic manipulation using

CRISPR/Cas9 and morpholino technology allowed efficient mapping of the novel *Cronos* internal promoter as well as testing of alternative mechanistic hypotheses for this phenomenon. Nonetheless, implicit in our work is the assumption that findings from zebrafish skeletal muscle sarcomere development in *ttna* homozygote mutants (albeit with the paralogous *ttnb* present for rescue) are relevant towards understanding disease development in humans with *TTN* mutations. Prominent differences between these systems must be noted. With the exception of certain childhood-onset forms of skeletal and cardiac myopathy arising from homozygous *TTN* truncations (*Carmignac et al., 2007*; *Ceyhan-Birsoy et al., 2013*), the mode of inheritance in *TTN* truncation mutation patients is heterozygous with marked variable expressivity and incomplete penetrance (*Herman et al., 2012*; *Roberts et al., 2015*; *Itoh-Satoh et al., 2002*; *Norton et al., 2013*; *van Spaendonck-Zwarts et al., 2014*). Moreover, disease onset usually occurs in adulthood. Such an inheritance pattern is typical of many of the familial forms of cardiomyopathy, and is believed to imply additional modifying effects, potentially from diverse sources as autoimmunity and viral infection as well as contributions from additional genetic variants (*Mrh et al., 2015*). Nonetheless, taking all of our results together, we find it a highly plausible explanation that a superimposed deficiency in expression of the *Cronos* protein product would result in the more severe forms of human cardiac disease seen in DCM patients with C-terminal *TTN* truncations.

With these limitations noted, our results have two clinical implications for titinopathy patients. Firstly, by precisely mapping the location of the alternate promoter to the e239-e240 intron in humans, we are in a better position to give mutation-specific guidance to TTN truncation patients regarding the potential severity of their disease, with the caveat that additional factors (genetic and environmental) also modulate phenotypic severity. Along these lines, given the overlapping distribution of mutations in unselected cases and controls, it is unlikely that highly accurate patient prognoses would be realized from genotype information alone (*Figure 7—figure supplement 1*), although there does appear to be nearly complete separation of mutation distribution for the more extreme phenotypes (end-stage DCM and senior athletes). Secondly, our work implies that strategies that augment activity of the internal TTN promoter and levels of the *Cronos* isoform might provide therapeutic benefit in TTN truncation patients, although it is unclear whether there is a restricted developmental window where this might be possible.

In addition to clinical ramifications, our work may shed light on a long-standing mystery regarding sarcomeric architecture. Electron microscopy reconstructions of serial sections of skeletal muscle dating back over 50 years presented a puzzle: the thin/thick filament arrangement in the sarcomere alters its geometry, progressing from a tetragonal lattice arrangement at the Z-band to a trigonal arrangement in the A-band (*Pringle, 1968*), with the transition taking place near the I-A junction (*Traeger et al., 1983*). The implication for TTN would be that there would be a resulting increase in the copies of TTN protein, with four at the Z-band and six at the A-band and M-line (*Liversage et al., 2001*), which is difficult to reconcile with the fact that a single TTN molecule spans the entire length, from Z-band to M-line. Our data provides a simple explanation for this observation, with the *Cronos* protein product selectively increasing copies of TTN at the distal I-band, A-band and M-line. Our work also proposes an alternative explanation for the consistently observed 'T2' band on protein gels, which migrates at the expected molecular weight of *Cronos*. This has invariably been labeled a C-terminal degradation product, as it reacts with A-band antibodies (*Opitz et al., 2004*). However, in keeping with *Cronos* expression patterns in development, the T2 band was prominently seen in developing mouse (*Lahmers et al., 2004*) and rat hearts (*Opitz et al., 2004*), but was markedly reduced in adulthood. It is possible that, at least in some situations, T2 and *Cronos* represent the same isoform.

These results also raise a number of questions related to the function of the *Cronos* isoform. Given its more marked expression in early development, is it important only for the initial sarcomerogenesis or in subsequent aspects of muscle function? Is it likely to be reactivated under stress? And how fast does it turn over compared to full-length TTN? Models with selective disruption and tagging of this isoform should begin to answer these questions. Finally, it is possible that unrecognized internal promoters influence disease severity in truncating mutations of other genes. It is likely a more nuanced diagnostic and prognostic interpretation of truncating mutations will be needed.

## Materials and methods

### sgRNA design, cloning, and in vitro transcription

sgRNA's were designed using an in-house *Python* script, which scans genomic sequences for 23 bp sequences with a terminal NGG (PAM sequence), and prioritizes a GG at the 5' end because of the use of T7 RNA polymerase for gRNA synthesis. All sgRNA's had only a single genome match. Oligo-nucleotides corresponding to the sgRNA sequence were annealed and then cloned into vector DR274 (Addgene plasmid 42250) as previously described (*Hwang et al., 2013*). For sgRNA synthesis, 100 ng of plasmid template was used in a T7 in vitro transcription reaction (MAXIscript T7, Life Technologies, Carlsbad, CA, AM1312M). RNA products were digested with DNAse I and purified by column (RNA Clean and Concentrator, Zymo Research, Irvine, CA, R1016).

### Cas9 induced genomic cleavage and genotyping

Recombinant Cas9 was prepared as previously described (*Jinek et al., 2012*). A mixture of recombinant Cas9 (0.5 ug/uL) and sgRNA 5-(40 ng/uL) was prepared and injected (1 nL total volume) into one-cell stage embryos (Ekwill strain). Cutting efficiency was assessed by pooling injected embryos at 24 hr post fertilization (hpf), extracting genomic DNA, amplifying the genomic targeted region by PCR (see *Supplementary file 1B* for primers), and using the Cel1 assay (*Oleykowski et al., 1998*) to detect evidence of targeted mutations.

To pinpoint the exon at which skeletal muscle disarray occurs, recombinant Cas9 and high dose gRNAs (100 ng/uL) corresponding to exons 113, 114, 115, 116, 122 and the predicted start codon for the alternative promoter were each injected into ~120 one-cell stage embryos. Given that homozygotes of either N- or C-terminal truncation mutants of *ttna* cause cessation of cardiac contraction, we focused our attention on those fish with obvious contractile defects and further phenotyped these through motility assays and immunohistochemistry (see below). For each of the targeted exons, 10 embryos with cessation of contraction were analyzed by immunostaining.

### Husbandry and genotyping

Zebrafish founders were outcrossed onto the Ekwill background, and subsequent progeny out-crossed again. F2 founders were then in-crossed to obtain F3 generation homozygote mutants, which were used for phenotypic and genomic analysis. Genotyping was performed using the Cel1 assay (*Oleykowski et al., 1998*) and/or allele-specific PCR analysis (*Gaudet et al., 2007*).

### Immunohistochemistry

Embryos were fixed and stained using standard procedures (*Deo et al., 2014*). Briefly, anesthetized embryos at 3 dpf were fixed in 4% paraformaldehyde (PFA) for 12 hr. Fixed tissue was washed in PBS + 0.1% Triton X-100 (PBST), permeabilized with 10 ug/mL proteinase K in PBST, and then refixed with 4% PFA for 20 min after which it was subjected to standard blocking and antibody incubations. Primary antibodies used were mouse anti-human alpha-actinin (Life Technologies, MA1-22863, 1:500 dilution) and titin T11 (Sigma-Aldrich, St. Louis, MO, T9030, 1:100) and T12 (1:10, a generous gift from Elisabeth Ehler and Dieter Fuerst). The secondary antibody was donkey-anti-mouse conjugated to an Alexa-488 fluorophore (Life Technologies, R37114, 1:500).

### Imaging and data analysis

Videos at 100–250 frame per second of beating hearts were taken from live 72 hr post-fertilization (hpf) embryos using an Axiozoom V16 Stereo (Carl Zeiss, Oberkochen, Germany) with a 10× objective lens and an Orca Flash 4.0T high-speed digital camera (Hamamatsu Photonics, Hamamatsu City, Shizuoka, Japan). Confocal images were collected on a Leica TCS SP5 with a 63x objective lens. To assess motility, embryos were tapped gently with a whisker.

### Morpholino knockdown

ATG- and splice-blocking morpholinos to *ttna* (conventional and alternative promoter) and *ttnb* were designed with the help of Gene Tools staff (http://www.gene-tools.com/). Morpholino sequences were TAA ATG TTG GAG CTT GCG TTG ACA T (ATG) and AAT TAC CTT TGA TTG TGA CTT TGG T (*Seeley et al., 2007*) for *ttnb* and CCT GTG TTG TCA TGG TGG AAG GCA C (conventional ATG),

GTT CCA GGA GAC ACA GGT AAT CCC T (internal ATG) and TGC AAC TGA TAC TCA CCT TCT CCA C (exon4-intron4 junction) for *ttna*. Morpholinos were re-suspended in sterile water to a concentration of 1 mM and diluted to working concentration in sterilized water. Male and female wild-type adult zebrafish were housed and embryos bred using standard protocols (*Westerfield, 1993*). Morpholinos were introduced into the zebrafish yolk via microinjection at no later than the two-cell developmental stage. Injected embryos were then kept at 28.5°C in E3 solution (5 mM NaCl, 0.17 mM KCl, 0.33 mM CaCl$_2$, 0.33 mM MgSO$_4$). The conventional promoter *ttna/ttnb* ATG-morpholinos were highly toxic, even at low concentrations. We thus focused on using the *ttna* and *ttnb* splice-blocking morpholino's as well as the internal promoter *ttna* ATG morpholino at working concentrations of 5 ng/nl. To further reduce toxicity, we co-injected the ttna splice-blocking morpholino with a previously described p53 morpholino at 5 ng/nl (*Robu et al., 2007*). In terms of sample size, we typically injected 100–150 embryos for every experiment.

## Zebrafish RNA-sequencing and analysis

High depth RNA-Sequencing of wild-type embryonic hearts was first performed to select constitutive exons for gRNA targeting. Embryo heart purification was carried out as described (*Geoffrey Burns and Macrae, 2006*). Briefly, 300–400 embryos of *Tg (cmcl2-GFP)* at 3 dpf were pooled for a single sequencing experiment. Embryo Dissociation Medium (EDM) was freshly prepared using Leibovitz's L-15 Medium (Life Technologies, 11415–064) containing 10% FBS (Sigma, F-2442), and kept ice-cold prior to use. After washes with EDM, embryos in 1 ml of EDM were drawn to a 3 ml syringe with 19G needle and expelled back to the tube for 30 times. Large embryo fragments were filtered out using a 100 um filter, and the filter was washed several times with more EDM. The flow-through was collected in a 60-mm glass petri dish on ice, and then applied to a 40 um filter. The filter was inverted and the retained fragments including hearts were washed off into a glass dish with 5x1ml EDM. GFP + hearts were sorted under fluorescent light using a Leica (Leica Camera, Wetzar, Germany) M205 FA fluorescence microscope. Trizol-extracted RNA was purified by column (RNA Clean and Concentrator, Zymo Research, R1016). Sequencing libraries were prepared using the SureSelect Strand Specific RNA-Seq kit (Agilent Technologies, Santa Clara, CA, G9691A), following the manufacturer's instructions. 101 bp paired end sequencing was performed on a HiSeq 2500 sequencer (Illumina, San Diego, CA).

For analysis of splicing of *ttna* mutants, hearts and skeletal muscle (trunk) were isolated from *ttna* mutant and wild-type zebrafish embryos (72 hrpf) and adults using manual dissection after euthanasia. Pooled hearts and skeletal muscle from 10 fish were used for subsequent library preparation. The tissue was lysed in Trizol (Life Technologies). RNA was recovered from the aqueous phase and purified purified by column (RNA Clean and Concentrator, Zymo Research, R1016) according to manufacturer's protocol. cDNA was prepared using the Ovation RNA-Seq System V2 (NuGEN Technologies) with sonication into 200bp fragments using an M220 sonicator (Covaris, Woburn, MA). Sequencing libraries were prepared using the Ovation Ultralow Ultralow DR Library System (NuGEN Technologies, San Carlos, CA). 101 bp single end sequencing was performed on a HiSeq 2500 sequencer in rapid run mode, with 7 samples per lane of the flow cell.

Reads were mapped to the *dr7* build of the zebrafish genome using TopHat (*Trapnell et al., 2009*). Samples were normalized using the edgeR package (*Robinson et al., 2010*), with trimmed mean of M values (TMM) normalization. Percent spliced in estimates were made using in-house Python scripts implementing previously published methods of splice isoform quantification using exon-exon junction reads (*Pervouchine et al., 2013*), which avoid the estimation difficulties arising from non-uniform read coverage of exons (*Kakaradov et al., 2012*).

## Mouse RNA-sequencing and analysis

The heart from an E12.5 129X1/SvJ x C57BL/6 cross mouse was disrupted in Trizol with plastic pestle and passed through QIAshredder (Qiagen, Hilden, Germany, 79654) for homogenization. RNA was isolated from Trizol using a 1:1 mixture with 70% ETOH followed by column purification and DNase treatment (RNeasy Mini kit, Qiagen, 74104). The cDNA was generated and amplified using the Ovation RNA-Seq System V2 (NuGEN Technologies) and the library was prepared using the Ovation Ultralow Ultralow DR Library System (NuGEN Technologies). 101 bp single end sequencing was

performed on a HiSeq 2500 sequencer. Reads were mapped to the *mm9* build of the mouse genome using TopHat (*Trapnell et al., 2009*).

## Nonsense-mediated decay rate estimates

To estimate nonsense-mediated decay (NMD) rates, we evaluated allelic imbalance in heterozygote embryos. For each mutant of interest, total RNA was extracted from three heterozygote embryos. Primers, designed to span at least one exon-exon junction, were used to amplify ~100 bp surrounding the mutation of interest. For each embryo, 2 independent PCR reactions - each with 10 amplification cycles - was performed. Next generation sequencing libraries were prepared using the NEBNext Ultra kit (New England Biolabs, Ipswich, MA, E7370S) with 10 cycles post-amplification. Samples were assessed by 100 bp single end sequencing on a MiSeq instrument (Illumina). The ratio of mutant to wild-type reads was used as a measure of allelic imbalance. For each mutation, between 600 and 10,000 reads were used for ratio estimates.

## Quantitative real-time PCR to assess ratio of Cronos to full-length ttna/Ttn in zebrafish and mouse

Forward primers mapping to either e115 or the Cronos isoform was used in combination with a reverse primer spanning the e116-e117 exon-exon junction. Similar primers were designed to detect *Cronos* vs. full-length Ttn levels in mouse. Standard curves using a cloned PCR product were used to compare relative primer efficiencies (data not shown). Quantitative real time-PCR analyses were carried out with cDNA templates using SensiFast SYBR green (Bioline Reagents, London, UK) on the CFX384 Real Time PCR system (BioRAD, Hercules, CA) (*Heredia et al., 2013*).

Zebrafish samples for qPCR consisted of embryonic and adult hearts and trunk muscle, and were isolated as described above. Mouse heart and skeletal muscle samples were taken from C57BL/6 or CD1 mice purchased from Jackson laboratory. All mice were euthanized by $CO_2$ inhalation and cervical dislocation prior to surgical dissection of relevant tissues.

## Human RNA-Sequencing analysis

We downloaded fastq format files for DCM patients and healthy controls corresponding to GEO Accessions GSM1380718, GSM1380719, GSM1380722, and GSM1380723. Reads were mapped to the *hg19* build of the human genome using TopHat (*Trapnell et al., 2009*). PSI values were computed as described above.

## Use of Publicly Available DNAse-Seq and ChIP-Seq data

We downloaded bed format files corresponding to GEO Accessions GSM665817 (fetal heart), GSM530661 (fetal heart), GSM665817 (fetal heart), GSM665824 (fetal heart), GSM665830 (fetal heart), GSM701533 (fetal trunk), GSM772735 (fetal heart), GSM906406 (left ventricle), GSM1027322 (pediatric heart), GSM1058781 (fetal leg), GSM1160200 (fetal trunk), and GSM530654 (fetal heart). Depth of coverage analysis was performed using in-house *R* scripts and data plotted using the ggplot2 package (*Wickham, 2009*).

## 5' RACE

5' RACE was performed using published protocols (*Picelli et al., 2014*; *Matz et al., 1999*). An oligonucleotide for template switching (TSO, Univ_UMI_RACE, *Supplementary file 1B*) was purchased from Exiqon (Woburn, MA). Total RNA was obtained from three sources. Zebrafish RNA was isolated from zebrafish skeletal muscle, as described above. Mouse RNA was isolated from cardiac tissue of a euthanized adult mouse (C57BL/6) using manual dissection of the heart followed by tissue homogenization, Trizol extraction, and RNA column purification (RNA Clean and Concentrator, Zymo Research, R1016). Fetal human heart (31 week old, male) RNA was purchased (BioChain Institute, Newark, CA, R1244122-50). First strand synthesis was performed using a TTN gene-specific primer (*Supplementary file 1B*) and the TSO, with SmartScribe reverse transcriptase (Clontech, Mountain View, CA, 639536) and with added SUPERase In RNAse inhibitor (Life Technologies, AM2694). Uracil-DNA Glycosylase (New England Biolabs, M0280S) was used to degrade any remaining TSO according to manufacturer's instructions. PCR amplification of the resulting cDNA was performed using a nested gene-specific primer (*Supplementary file 1B*) and a universal 5' primer

(*Supplementary file 1B*) with a touch-down annealing temperature protocol. PCR products were gel purified (Zymo Research, D4001), blunt-end cloned using the pGEM-T Easy Vector system (Promega, Madison, WI, A12360) and analyzed by Sanger sequencing.

## In situ hybridization

Whole-mount in situ hybridization with digoxigenin-labeled mRNA antisense probes to *Cronos* and *ttna* was performed as previously described (*Thisse and Thisse, 2008*). For *Cronos*, an 89nt digoxigenin-labeled riboprobe unique to this isoform was hybridized at 60°C. For *ttna*, a 453nt riboprobe corresponding to a constitutive region was hybridized at 68°C. In both cases embryos were developed in BM purple (11442074001, Sigma-Aldrich).

## Protein electrophoresis

Briefly, euthanized zebrafish embryos were Dounce homogenized in lysis buffer (500 mM Tris pH 7.4 150 mmM NaCl, 1% NP-40, 1 mM PMSF, 1x protease inhibitor cocktail [Sigma-Aldrich]) on ice and centrifuged at 14,000 g for 10 min at 4C. The denaturing/loading buffer (8 M urea, 2 M thiourea, 3% SDS, 75 mM DTT, 0.1% bromophenol blue, 0.05% Tris-HCl pH 6.8) was mixed with the lysate at a 4:1 ratio and the resulting sample denatured at 60°C for 15 min. Electrophoresis was performed using a denaturing 1% agarose gel at 15mA for 3–6 hr (3 hr for Myosin and 6 hr for Titin) on a Hoefer (Holliston, MA) SE 600 gel unit (*Steffen, et al., 2007*; *Warren, et al., 2003*). Gels were fixed for at least 1 hr after running, briefly washed in water and vacuum dried at 40C overnight. The following day, the gel was rehydrated and stained with SYPRO Ruby protein gel stain (S4942, Sigma-Aldrich) according to manufacturer's instructions.

## Senior athlete recruitment

199 healthy senior athletes with no prior history of cardiac disease were recruited between 2004 and 2012 at the Huntsman World Senior Games (held in St George, Utah). DNA from whole blood was extracted using the Gentra Puregene DNA isolation kit (Qiagen, 158445).

## Targeted capture-based sequencing and variant calling

Targeted capture and variant calling were performed as previously described (*Deo, et al., 2014*). Briefly, NimbleGen's SeqCap EZ Choice technology was used to construct oligonucleotide probes complementary to the exons (Ensembl GRCh37) of 160 genes involved or predicted to be involved in Mendelian forms of cardiac disease (*Supplementary file 1A*). Genomic DNA libraries were sheared by sonication on a M220 sonicator (Covaris), hybrdized to the capture probes, and libraries prepared from the captured material using the Kapa Biosystems (Wilmington, MA) Library Preparation Kit. Libraries were sequenced on an Illumina HiSeq 2500 sequencer with multiplexing of 24 samples per lane. Assembly was performed using *bwa* (*Li and Durbin, 2009*) and variants called using the *GATK* (*McKenna, et al., 2010*). Median coverage was 455-fold with 99.6% of exons found to be callable for the major cardiac N2BA isoform (ENST00000591111)and 98.4% callable for the Inferred Complete isoform (ENST00000589042). Seven samples were captured with a prior cardiomyopathy gene panel of 116 genes (*Deo, et al., 2014*).

Coding sequence mutations resulting in a premature nonsense codon or frame-shift were considered to be truncating mutations. For mutations of canonical (-1/-2) splice acceptor variants, which would most likely result in exon skipping, we further required that the downstream exon was not in the same codon phase as the skipped exon, as this would fail to produce a frame-shift and thus not result in a truncation (such a situation is common in the I-band). Likewise, for mutations of canonical splice donor variants (+1/+2), which we predicted would result in intron read-through, we performed in silico translation of the resulting retained intron to ensure that a premature truncation was observed (this was not always the case, especially for the I-band).

## Statistical analysis

All statistical analysis was performed in R (3.1.1). A difference in proportion test was applied to the prevalence of TTN truncations in senior athletes vs. literature controls, as well as the fraction of TTN truncations that map to the Novex-3 exon.

# Additional information

## Funding

| Funder | Grant reference number | Author |
|---|---|---|
| American Heart Association | 15POST25090054 | Jun Zou |
| National Heart, Lung, and Blood Institute | DP2 HL123228 | Rahul C Deo |
| National Heart, Lung, and Blood Institute | U01 HL107440-03 | Rahul C Deo |
| National Heart, Lung, and Blood Institute | K08 HL093861 | Rahul C Deo |

The funders had no role in study design, data collection and interpretation, or the decision to submit the work for publication.

## Author contributions

JZ, Conception and design, Acquisition of data, Analysis and interpretation of data, Drafting or revising the article; DT, AP, AnP, CY, CS, SP, EH, JG, RW, Acquisition of data, Analysis and interpretation of data; MB, LFT, Conception and design, Acquisition of data, Analysis and interpretation of data; EWT, Analysis and interpretation of data, Drafting or revising the article; IM, JL, DOH, Acquisition of data, Contributed unpublished essential data or reagents; ChY, PYK, Conception and design, Acquisition of data; LDW, Acquisition of data, Drafting or revising the article; RLH, CRP, Drafting or revising the article, Contributed unpublished essential data or reagents; SRC, Conception and design, Acquisition of data, Drafting or revising the article; MJ, JPK, Conception and design, Contributed unpublished essential data or reagents; RCD, Conception and design, Analysis and interpretation of data, Drafting or revising the article

## Author ORCIDs

Erron W Titus, http://orcid.org/0000-0001-6868-9121
Ronald L Hager, http://orcid.org/0000-0002-5128-4635

## Ethics

Human subjects: Human genetic studies were performed according to institutional guidelines and with the full approval of the University of California San Francisco Committee on Human Research (CHR#10-00207) and all studies performed were in keeping with the original informed consent forms. Informed consent and consent to publish was obtained from all participants.
Animal experimentation: All zebrafish and mouse experimental work conformed to the 'Guide for the Care and Use of Laboratory Animals' published by the US National Institutes of Health (NIH Publication No. 85-23, revised 1996). Animal work was performed according to institutional guidelines with the full approval of the University of California Institutional Animal Care and Use Committee (protocols AN090013-03 and AN107039-01).

# Additional files

## Supplementary files

• Supplementary file 1. (A) 160 established and predicted cardiomyopathy and/or channelopathy genes whose exomes were targeted by capture-based sequencing; (B) Primers used in this study.

## Major datasets

The following previously published datasets were used:

| Author(s) | Year | Dataset title | Dataset URL | Database, license, and accessibility information |
|---|---|---|---|---|
| Liu Y, Morley MP, Brandimarto J, Hannenhalli S, Hu Y, Ashley EA, Tang WH, Moravec CS, Margulies KB, Cappola T, Li M | 2015 | RNA-Seq Identifies Novel Myocardial Gene Expression Signatures of Heart Failure [RNA-seq] | http://www.ncbi.nlm.nih.gov/geo/query/acc.cgi?acc=GSE57344 | Publicly available at the NCBI Gene Expression Omnibus (Accession no: GSE57344). |
| | | Stamatoyannopoulos J | 2010 | University of Washington Human Reference Epigenome Mapping Project |
| http://www.ncbi.nlm.nih.gov/geo/query/acc.cgi?acc=GSE18927 | | Publicly available at the NCBI Gene Expression Omnibus (Accession no: GSE18927). | | |

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
