## [Decision Letter]

Thank you for submitting your work entitled "An internal promoter underlies the difference in disease severity between N- and C-terminal truncations of Titin in zebrafish" for peer review at *eLife*. Your submission has been evaluated by Mark McCarthy (Senior Editor) and three reviewers, one of whom, Harry Dietz, is a member of our Board of Reviewing Editors. Your manuscript has generated considerable enthusiasm. The reviewers have discussed the reviews with one another and the Reviewing Editor has drafted this decision to help you prepare a revised submission.

This interesting manuscript provides a provocative explanation for a well-described phenotype-genotype correlation in dilated cardiomyopathy (DCM) caused by mutations in the gene encoding titin (TTN). Specifically, C-terminal heterozygous truncating mutations associate with a much more severe presentation of DCM than do N-terminal truncating mutations, with an apparent discriminating boundary between exons 115 and 116. In this manuscript, the authors recapitulate this experience using a series of targeted mutations in the *ttna* gene in zebrafish, which also have a paralogous *ttnb* gene. Evidence is provided to suggest that these observations are not based upon alternative splicing, differential efficiency of nonsense-mediated mRNA decay (NMD) or differential dominant-negative potential of the predicted truncated protein products derived from the mutant alleles. Instead, the data strongly suggest the relevance of an alternative promoter that lies between exons 115 and 116, giving rise to an mRNA and protein (termed *Cronos*) that is expressed (and putatively functional) with proximal but not distal premature termination codon mutations. Overall, this is a very elegant study with quality data and largely convincing conclusions. However, there are a number of issues and questions that need to be addressed before we can consider this further.

1) The most substantive concern expressed by multiple reviewers relates to the relative strengths and limitations of the zebrafish model. The lack of correlation between location of truncating mutations and severity of cardiomyopathy in this model is inadequately explored and discussed. A comprehensive tissue-specific quantitative and developmental survey of alternative promoter utilization in both heart and skeletal muscle is indicated. The rationale for studying homozygous variation in zebrafish is somewhat difficult to justify completely and has an impact on the generalizability of the results. The authors should consider higher resolution phenotyping of heterozygous animals with or without environmental stressors. They should also comprehensively describe (and perhaps illustrate) what is known regarding species-specific TTN genomic structure and message biology.

2) NMD is generally a very efficient process. There is generally an apparent polarization effect, with greater efficiency for proximal PTCs, with the potential for less efficiency for distal PTCs and near-universal loss of NMD for PTCs in the very distal portion of the penultimate exon or in the last exon. In this light, it is surprising that all assayed PTCs (none in last exons) had a relatively modest effect on transcript abundance (∼50% of WT). The details regarding how this assay was performed are minimal and should be expanded. Would the assay detect the *Cronos* transcript that is expected to be present for N- but not C-terminal truncations? If so, why is this not reflected in the results of this experiment?

3) Additional experimental details and/or controls are also needed for the dominant-negative assay, for example the control for efficient morpholino-induced exon skipping related to observation of cardiomyopathy in WT mice. There needs to be some demonstration of high efficiency in skeletal muscle. Also, the authors should consider the possibility of a dosage-dependent dominant-negative effect when interpreting their results.

4) The apparent demonstration of the *Cronos* protein product (Figure 5) is not convincing. There is poor resolution, no demonstration of specificity, no identification of full-length protein, no explanation for the variation in intensity of other protein isoforms that vary between genotypes, and no loading control.

5) Much more detail is needed regarding the exact location and predicted functional consequence of the truncating mutations observed in senior athletes vs. various control populations. When compared to healthy volunteers or participants in heart study cohorts (numbers not specified), senior athletes had about the same frequency (∼1.0-1.5%) of TTN truncation mutations. However, all senior athletes had their truncating mutation in the very distal portion of the Novex-3 isoform (that is rather poorly described in this manuscript), while the other groups largely showed ‘other’ types of truncations. Apparently, there is an alternative TTN transcript that includes a large (∼6.5kb) terminal exon that also contains a 3' cleavage and polyadenylation signal. By report, the Novex-3 transcript contains all proximal exons, and terminates considerably short of the alternative promoter described here. Are the truncations in the very distal portion of the Novex-3 exon seen in robust old age because they have no functional impact at all (i.e. normal levels of both full-length and Novex-3 message and protein due to avoidance of NMD in both)? While the data might even suggest a selective advantage, there is no mechanism that can even be inferred by the data and conclusions of this manuscript. Perhaps more importantly, what was the specific nature of the ‘other’ truncating mutations in the healthy volunteers and other unselected cohorts? Were they greatly enriched for N- as opposed to C-terminal truncations (defined by the exon 115/116 boundary), as might be predicted by this study? Were the Novex-3 truncations seen in the FHS and JHS cohorts also in the extreme C-terminal exon, or were they more widely distributed?

6) There are concerns regarding generalizations made about existing TTN phenotype-genotype correlations in human cardiomyopathy patients. The initial inference that C-terminal titin truncation variants are uniformly classified as pathogenic mutations is incorrect. There is a substantial excess of C-terminal truncating variants in DCM subjects, but there are also such variants in the control groups so the position of a variant is insufficient (at least at this resolution) to define pathogenicity. This observation also raises important questions regarding the mechanisms of pathogenicity. In addition, it is worth noting that there are remarkably few of the proposed truncating mutations that have been demonstrated to result in truncated protein and there are only two extant families where there are any segregation data of significance between titin variants of any sort and the DCM phenotype. In each case, the variant is not clearly truncating when studied closely. Finally, the spectrum of variation in titin that leads to skeletal muscle involvement is not so systematically described but many DCM subjects (with heterozygous genotypes) have subclinical skeletal muscle involvement when examined rigorously. The suggestion that TTN mutations cause HCM is controversial and should be stated as such. The distinction between N- and C- terminal truncating mutations with regard to DCM is not as discrete and robust as is suggested by the authors, with documentation of cases with N-terminal truncations (i.e. Herman DS et al, NEJM 2012). Also, given the complexities of TTN transcript biology that remain to be fully explored in fish, mice and especially people, the authors’ suggestion that it is now possible to definitively classify C-terminal variation in the titin gene is overstated. While this work will undoubtedly contribute to progress, much more needs to be done to reliably inform patient management and counseling.

[Editors' note: further revisions were requested prior to acceptance, as described below.]

Thank you for resubmitting your work entitled "An internal promoter underlies the difference in disease severity between N- and C-terminal truncations of Titin in zebrafish" for further consideration at *eLife*. Your revised article has been favorably evaluated by Mark McCarthy (Senior Editor), Harry Dietz (Reviewing Editor), and two reviewers. The manuscript has been improved but there are some remaining issues that need to be addressed before acceptance, as outlined below:

Please take note of the additional comments by reviewer #3 and specifically further highlight the complexity and therefore inherent limitations of the zebrafish system.

Reviewer #2:

This revised manuscript addresses all of my prior comments well. The manuscript is substantially improved. I have no additional concerns or recommendations.

Reviewer #3:

The authors suggest that they have clarified the relationship between zebrafish and human titin biology, but they have simply fitted their observations to a model where homozygous titin null zebrafish skeletal muscle is representative of human heterozygous null cardiac muscle. While this may be the case, the fact is that sarcomerogenesis itself is disrupted during development in the fish limb of this construct, while humans with the proposed orthologous trait have no demonstrable structural defect in the sarcomere and survive to middle age.

The authors' also assert that components of the observed disparities between fish and human hearts and fish and human skeletal muscle are a consequence of the associated levels of *ttnb* and *Cronos* in the different tissues in each species. It would be important to formally demonstrate these assertions with rescue experiments in the different tissues, or as noted in prior reviews, to consider and test alternative explanations including the presence or absence of external stressors. The authors did test two rather acute and rather similar perturbations without success, though the zebrafish would seem to be amenable to much more extensive and stringent challenge.

The RNAseq data more robustly quantitate the levels of NMD and are reassuring, though the distribution along the transcript is somewhat discordant with prior observations in other genes.

I remain somewhat unsure how to interpret the failure of *ttna* knockdown to rescue the *ttna* homozygous mutants. It is difficult to see how a mechanism that may be dependent on the presence of a threshold amount of wild-type (WT) protein, can be excluded by the knockdown of both mutant alleles. This serves only to emphasize the failure of the zebrafish experiments as described to capture the biology of either human skeletal or cardiac titin disease.

The authors have revised their statements on the inferences of their work for clinicians and in doing so have created an improved and carefully nuanced description of the field.

---

## [Author Response]

*This interesting manuscript provides a provocative explanation for a well-described phenotype-genotype correlation in dilated cardiomyopathy (DCM) caused by mutations in the gene encoding titin (TTN). Specifically, C-terminal heterozygous truncating mutations associate with a much more severe presentation of DCM than do N-terminal truncating mutations, with an apparent discriminating boundary between exons 115 and 116. In this manuscript, the authors recapitulate this experience using a series of targeted mutations in the* ttna *gene in zebrafish, which also have a paralogous* ttnb *gene. Evidence is provided to suggest that these observations are not based upon alternative splicing, differential efficiency of nonsense-mediated mRNA decay (NMD) or differential dominant-negative potential of the predicted truncated protein products derived from the mutant alleles. Instead, the data strongly suggest the relevance of an alternative promoter that lies between exons 115 and 116, giving rise to an mRNA and protein (termed* Cronos*) that is expressed (and putatively functional) with proximal but not distal premature termination codon mutations. Overall, this is a very elegant study with quality data and largely convincing conclusions. However, there are a number of issues and questions that need to be addressed before we can consider this further. 1) The most substantive concern expressed by multiple reviewers relates to the relative strengths and limitations of the zebrafish model. The lack of correlation between location of truncating mutations and severity of cardiomyopathy in this model is inadequately explored and discussed. A comprehensive tissue-specific quantitative and developmental survey of alternative promoter utilization in both heart and skeletal muscle is indicated. The rationale for studying homozygous variation in zebrafish is somewhat difficult to justify completely and has an impact on the generalizability of the results. The authors should consider higher resolution phenotyping of heterozygous animals with or without environmental stressors. They should also comprehensively describe (and perhaps illustrate) what is known regarding species-specific TTN genomic structure and message biology.*

With regard to the primary area of contention, we recognize that we have done an inadequate job drawing a path between our observations in the zebrafish model and human cardiac disease. In the revised manuscript we have sought to make a clearer, more compelling case with the following changes:

A) Comparison of zebrafish and human titin:

The following text is included verbatim in the revised manuscript and we have included a new Figure (Figure 1—figure supplement 1) to graphically illustrate the domain organization of human TTN and zebrafish *ttna* and *ttnb*, as suggested. Our primary objectives in this section are to 1) highlight the similarities between the human and zebrafish Titin proteins, and 2) emphasize that the paralogous ttna and ttnb proteins are both essential for skeletal muscle sarcomere development, creating a scenario where, at least in skeletal muscle, studying a homozygote knockout of one of these genes bears resemblance to studying a heterozygote in mammals.

We have also rephrased the text accordingly. Please see: “As a result of an ancestral genome duplication event, zebrafish include two titin genes: *ttna* and *ttnb* […] severe disruption of skeletal muscle sarcomeric architecture, highlighting the mutual contribution of the two genes (14).”

B) Tissue-specific quantitative and developmental survey of *Cronos* expression:

In the revised manuscript, we have added 3 new figure panels that characterize *Cronos* expression in zebrafish and mouse by whole mount in situ hybridization (with one probe specific to *Cronos*) and by quantitative real-time PCR (Figure 6). Our primary observation is that *Cronos* expression is at high levels in zebrafish skeletal muscle during development (∼2:1 ratio of *Cronos* to full-length *ttna*), but drops off dramatically during adulthood (∼1:70 ratio). Although the *Cronos* expression in zebrafish heart also appears to be developmentally regulated, it is much lower than in skeletal muscle, with a ∼1:5 ratio at 72 hr and 1:30 ratio in adulthood.

In contrast, the relative levels of *Cronos* in mouse heart are much higher than in skeletal muscle, with nearly a 1:1 ratio of *Cronos* to full-length Titin in E12.5 heart. Thereafter mouse cardiac expression of *Cronos* decreases to ∼20–30% of full-length Titin. Skeletal muscle expression varies widely across development and in different muscle beds with a 1:5 ratio in mouse hindlimb at P2, and between 1:16 and ∼1:1000 ratios in adult extensor digitorum longus (fast-twitch) and soleus (slow-twitch) muscle, respectively.

We indicate that the poor expression of *Cronos* in zebrafish heart can be a possible explanation for the indistinguishable cardiac phenotypes of N- and C-terminal truncation mutants, though the fact that *ttnb* plays a minimal role in cardiac development is probably also contributing to the severity of the *ttna* mutants.

In addition to the additional Figure 6, we have included two paragraphs summarizing the tissue-specific and developmental stage results: “We next surveyed the expression levels of *Cronos* in cardiac and skeletal muscle […] expression varies across development and in different muscle beds with a 1:5 ratio in mouse hindlimb at P2, and between 1:16 and ∼1:1000 ratios in adult extensor digitorum longus (fast-twitch) and soleus (slow-twitch) muscle, respectively.”

C) Interpretation of lack of variation in truncation mutation severity in heart:

We attribute the stark differences between skeletal muscle phenotypes in *ttna* N- and C-terminal truncation mutants to 1) the prominent expression of *Cronos* in developing skeletal muscle and 2) the presence of an important functional paralog, *ttnb*, which can substitute for *ttna* function. Unfortunately, in cardiac muscle the contribution of both of these factors is severely diminished, including low expression of *Cronos*, and a more minimal role for *ttnb* in cardiac sarcomerogenesis.

We are reassured that there is prominent expression of *Cronos* in developing mouse heart, which is in keeping with *Cronos* playing an important role in mammalian heart development and therefore consistent with the lack of *Cronos* intensifying DCM phenotypic severity. Interestingly, others have described a prominent band on protein gels of developing mouse and rat heart tissue at the expected molecular weight of *Cronos*, which is substantially diminished in adulthood. This band has been dismissed as a degradation product – but we speculate that in some cases this is very likely to be *Cronos*.

We touch on the limited cardiac expression of *Cronos* in zebrafish as well as the limited contribution of *ttnb* in heart development as explanation for our zebrafish cardiac results. We also include an additional paragraph in the Discussion: “Our work also proposes an alternative explanation for the consistently observed ‘T2’ band on protein gels, which migrates at the expected molecular weight of *Cronos*. This has invariably been labeled a C-terminal degradation product, as it reacts with A-band antibodies (25). However, in keeping with *Cronos* expression patterns in development, the T2 band was prominently seen in developing mouse (25) and rat hearts (26), but was markedly reduced in adulthood. It is possible that, at least in some situations, T2 and *Cronos* represent the same isoform.”

D) Perturbation induced discrimination of N- and C-terminal heterozygotes:

We agree that perturbation experiments in heterozygotes (such as was done with *Ttn^+/-^* mice, Gramlich et al, 2009) would have been more consistent with the human disease scenario, although the presence of paralogs complicates this situation in zebrafish. Although we tried isoproterenol (and angiotensin II) at several doses and at several developmental time points and used natriuretic peptide levels as readout, we were not able to consistently distinguish WT fish from heterozygotes (N- or C). Given the low cardiac expression of *Cronos*, we suspect that even with an optimized assay, the distinction of N- and C-terminal truncation mutants from one another would be very challenging. *2) NMD is generally a very efficient process. There is generally an apparent polarization effect, with greater efficiency for proximal PTCs, with the potential for less efficiency for distal PTCs and near-universal loss of NMD for PTCs in the very distal portion of the penultimate exon or in the last exon. In this light, it is surprising that all assayed PTCs (none in last exons) had a relatively modest effect on transcript abundance (∼50% of WT). The details regarding how this assay was performed are minimal and should be expanded. Would the assay detect the* Cronos *transcript that is expected to be present for N- but not C-terminal truncations? If so, why is this not reflected in the results of this experiment?*

This is an excellent point. Our prior approach to NMD estimate was inadequate and involved normalizing total *ttna* to *ttnb* levels in homozygote *ttna* null mutants. In the revised manuscript we have performed targeted RNA-Seq of the mutant exons in heterozygote embryos to estimate NMD by looking at allelic imbalance (Figure 2). Comparing raw counts of mutant and WT alleles, we see ∼80% of the mutant transcript degraded, in keeping with established NMD efficacy. Interestingly, we see very little variation across the length of the transcript. We have also performed qPCR to estimate the ratio of *Cronos* to full-length *ttna* in N-terminal and C-terminal mutants and find the expected result (Figure 5): a high ratio of *Cronos* to full-length *ttna* for all N-terminal mutants and a much lower rate, more comparable to WT, for the C-terminal mutants. We suspect that some variation in promoter activity for the full-length and *Cronos* transcripts could account for some of the variation in these ratios within the N- and C-terminal mutant groups.

We have modified the text and figures to include this data, which we believe strengthens our story. For example, we now state: “Given that NMD substantially reduces the levels of *ttna* transcripts with premature termination codons (Figure 2), we reasoned that we should be able to distinguish N- and C-terminal truncation mutants by the ratio of *Cronos* to full-length transcript. In *ttnan/n* mutants, which would degrade full-length *ttna* but not *Cronos*, we would anticipate a higher *Cronos*:full-length ratio than in *ttna*^c/c^mutants mutants, which should degrade both transcripts approximately equally. As expected, we see a much higher ratio of *Cronos*:full-length transcript levels in *ttna*^n/n^mutants than in *ttna*^c/c^mutants (Figure 5).” *3) Additional experimental details and/or controls are also needed for the dominant-negative assay, for example the control for efficient morpholino-induced exon skipping related to observation of cardiomyopathy in WT mice. There needs to be some demonstration of high efficiency in skeletal muscle. Also, the authors should consider the possibility of a dosage-dependent dominant-negative effect when interpreting their results.*

We have now performed qPCR to demonstrate ∼78% knockdown efficiency using our *ttna* morpholino. Either exon skipping or intron retention arising from morpholino injection should result in an N-terminal truncation on top of the C-terminal mutation. In conjunction with NMD, this intervention should result in residual C-terminal truncated protein being at ∼5% of the level of *ttnb*. Given that we see no amelioration in skeletal muscle function (fish remain immobile) or architecture, it is difficult to reconcile this observation with a dominant negative mechanism. Similarly, for dosage to be responsible for the difference between N- and C-terminal truncations, we would not expect to have seen similar levels of NMD for N- and C-terminal truncations nor see persistence of skeletal muscle pathology with such a low level of mutant C-terminal protein present.

In the manuscript, we have made the following changes: “Knockdown by morpholino was effective (∼78% knockdown efficiency, data not shown) and the combined action of NMD and morpholino knockdown would be expected to reduce mutant ttna protein to only ∼5% of the level of ttnb protein. Nonetheless we were unable to rescue skeletal muscle architecture in *ttna*^c/c^mutants (Figure 3). We thus concluded that *ttna*^c/c^ mutants do not act as dominant negatives and that an alternative explanation was needed.”

*4) The apparent demonstration of the* Cronos *protein product (Figure 5) is not convincing. There is poor resolution, no demonstration of specificity, no identification of full-length protein, no explanation for the variation in intensity of other protein isoforms that vary between genotypes, and no loading control.*

We have updated this gel to include morpholino knockdown controls as well as loading controls. We have found this gel technically challenging to execute, potentially because of the long running time ∼6 required, but we believe the results are now convincing.

*5) Much more detail is needed regarding the exact location and predicted functional consequence of the truncating mutations observed in senior athletes vs. various control populations. When compared to healthy volunteers or participants in heart study cohorts (numbers not specified), senior athletes had about the same frequency (∼1.0–1.5%) of TTN truncation mutations. However, all senior athletes had their truncating mutation in the very distal portion of the Novex-3 isoform (that is rather poorly described in this manuscript), while the other groups largely showed ‘other’ types of truncations. Apparently, there is an alternative TTN transcript that includes a large (∼6.5 kb) terminal exon that also contains a 3' cleavage and polyadenylation signal. By report, the Novex-3 transcript contains all proximal exons, and terminates considerably short of the alternative promoter described here. Are the truncations in the very distal portion of the Novex-3 exon seen in robust old age because they have no functional impact at all (i.e. normal levels of both full-length and Novex-3 message and protein due to avoidance of NMD in both)? While the data might even suggest a selective advantage, there is no mechanism that can even be inferred by the data and conclusions of this manuscript. Perhaps more importantly, what was the specific nature of the ‘other’ truncating mutations in the healthy volunteers and other unselected cohorts? Were they greatly enriched for N- as opposed to C-terminal truncations (defined by the exon 115/116 boundary), as might be predicted by this study? Were the Novex-3 truncations seen in the FHS and JHS cohorts also in the extreme C-terminal exon, or were they more widely distributed?*

We have modified this section and included 2 new figure panels: 1) one that depicts the position of mutations in DCM patients and control populations or from the senior athletes onto a schematic of the TTN meta-transcript 2) a ratio of mutations found in regions C-terminal to the *Cronos* promoter to those in the N-terminus in different populations. We have eliminated the figure panel that emphasizes the location of senior athlete mutations to the terminal exon of the Novex-3 isoform. In fact, when we referred to the Novex-3 isoform we meant only those mutations mapping to the terminal exon, which is unique to this isoform.

The following text is now included in the revised manuscript: “We compared the frequency and distribution of TTN truncation mutations to those found in unselected controls and healthy volunteers from a recent sequencing study (CTLlit) (7) […] As expected, in end-stage DCM we observed a much higher rate of mutants C-terminal to *Cronos* than N-terminal (30:1), followed by unselected DCM (37:11), literature controls (6:11, 2:2, 10:7 for FHS, healthy volunteers and JHS) and senior athletes (0:3).” *6) There are concerns regarding generalizations made about existing TTN phenotype-genotype correlations in human cardiomyopathy patients. The initial inference that C-terminal titin truncation variants are uniformly classified as pathogenic mutations is incorrect. There is a substantial excess of C-terminal truncating variants in DCM subjects, but there are also such variants in the control groups so the position of a variant is insufficient (at least at this resolution) to define pathogenicity. This observation also raises important questions regarding the mechanisms of pathogenicity. In addition, it is worth noting that there are remarkably few of the proposed truncating mutations that have been demonstrated to result in truncated protein and there are only two extant families where there are any segregation data of significance between titin variants of any sort and the DCM phenotype. In each case, the variant is not clearly truncating when studied closely. Finally, the spectrum of variation in titin that leads to skeletal muscle involvement is not so systematically described but many DCM subjects (with heterozygous genotypes) have subclinical skeletal muscle involvement when examined rigorously. The suggestion that TTN mutations cause HCM is controversial and should be stated as such. The distinction between N- and C- terminal truncating mutations with regard to DCM is not as discrete and robust as is suggested by the authors, with documentation of cases with N-terminal trucations (i.e. Herman DS et al, NEJM 2012). Also, given the complexities of TTN transcript biology that remain to be fully explored in fish, mice and especially people, the authors' suggestion that it is now possible to definitively classify C-terminal variation in the titin gene is overstated. While this work will undoubtedly contribute to progress, much more needs to be done to reliably inform patient management and counseling.*

We apologize for creating the impression that our work has resulted in newfound certainty in genotype-phenotype correlations. From our experience taking care of many dozens of families with inherited cardiomyopathies, such determinism is never the case. We have moderated language that makes any such suggestion in the revision. We would like to address some of the above comments and concerns:

1) The presence of ‘controls’ with C-terminal truncations does not eliminate the possibility that these mutations do in fact have cardiac consequences that fall short of overt heart failure, which was the motivation for studying a hyper-normal senior athlete population. Finding C-terminal truncations in the senior athlete population would have cast greater doubt on whether such mutations are sufficient to compromise cardiovascular fitness.

2) We are encouraged by this report of subclinical skeletal manifestations in DCM TTN patients though we cannot find any published studies that have rigorously documented this.

3) We have eliminated mention of the HCM association with TTN.

4) Our work actually does not address if N-terminal truncations – especially in constitutive exons – are deleterious. We analyzed the senior athletes with the hope that we might find N-terminal truncations though small sample size precludes robust conclusions. We do note that in Supplementary Table 12 of Herman et al, the dozen or so families with evidence of segregation in 2 or more members all had C-terminal truncations.

Please see the following passage in the manuscript: “Our results have two clinical implications for titinopathy patients […] although it is unclear whether there is a restricted developmental window where this might be possible.”

[Editors' note: further revisions were requested prior to acceptance, as described below.]

*Thank you for resubmitting your work entitled "An internal promoter underlies the difference in disease severity between N- and C-terminal truncations of Titin" for further consideration at* eLife*. Your revised article has been favorably evaluated by Mark McCarthy (Senior Editor), Harry Dietz (Reviewing Editor), and two reviewers. The manuscript has been improved but there are some remaining issues that need to be addressed before acceptance, as outlined below: Please take note of the additional comments by reviewer #3 and specifically further highlight the complexity and therefore inherent limitations of the zebrafish system.*Reviewer #2: *This revised manuscript addresses all of my prior comments well. The manuscript is substantially improved. I have no additional concerns or recommendations.*Reviewer #3: *The authors suggest that they have clarified the relationship between zebrafish and human titin biology, but they have simply fitted their observations to a model where homozygous titin null zebrafish skeletal muscle is representative of human heterozygous null cardiac muscle. While this may be the case, the fact is that sarcomerogenesis itself is disrupted during development in the fish limb of this construct, while humans with the proposed orthologous trait have no demonstrable structural defect in the sarcomere and survive to middle age. The authors' also assert that components of the observed disparities between fish and human hearts and fish and human skeletal muscle is a consequence of the associated levels of* ttnb *and* Cronos *in the different tissues in each species. It would be important to formally demonstrate these assertions with rescue experiments in the different tissues, or as noted in prior reviews, to consider and test alternative explanations including the presence or absence of external stressors. The authors did test two rather acute and rather similar perturbations without success, though the zebrafish would seem to be amenable to much more extensive and stringent challenge. The RNAseq data more robustly quantitate the levels of NMD and are reassuring, though the distribution along the transcript is somewhat discordant with prior observations in other genes. I remain somewhat unsure how to interpret the failure of* ttna *knockdown to rescue the* ttna *homozygous mutants. It is difficult to see how a mechanism that may be dependent on the presence of a threshold amount of wild-type (WT) protein, can be excluded by the knockdown of both mutant alleles. This serves only to emphasize the failure of the zebrafish experiments as described to capture the biology of either human skeletal or cardiac titin disease. The authors have revised their statements on the inferences of their work for clinicians and in doing so have created an improved and carefully nuanced description of the field.*

In the revised manuscript we have added two paragraphs (shown below) at the beginning of the Discussion to summarize our findings and highlight the strengths and limitations of the system we have used. Although we firmly believe that our findings represent a substantial advance in this field, with strong biological and clinical implications, we recognize the challenges of studying adult onset autosomal dominant diseases with incomplete penetrance in any sort of animal model. Even if one can bring about phenotypic distinctions with artificial perturbations or genetic modifiers or use increasingly sensitive phenotyping methods to capture differences, one cannot be sure such stresses or distinctions are in fact relevant to the human disease process. Furthermore, TTN brings about an even greater challenge because of the technical challenge of performing rescue experiments with a 110 kilobase transgene.

With regard to Reviewer 3’s final comment, we do not understand the statement of how a knockdown of a mutant allele in the setting of a WT alternative (*ttnb*) is of no value in assessing a dominant negative mode of action and instead “emphasize(s) the failure of the zebrafish experiments as described to capture the biology of either human skeletal or cardiac titin disease”. If one accepts that *ttnb* is rescuing the *ttna* null, this is analogous to performing allele specific knockdown of a mutant protein in a heterozygote and looking at phenotypic impact. In the manuscript, we stress: “Loss of function experiments across a number of systems have demonstrated Titin’s importance in diverse aspects of sarcomere development and function, including the ordered assembly of sarcomeric constituents […] Nonetheless, taking all of our results together, we find it a highly plausible explanation that a superimposed deficiency in expression of the *Cronos* protein product would result in the more severe forms of human cardiac disease seen in DCM patients with C-terminal TTN truncations.”